# Compensation for X-linked *Pdha1* silencing by *Pdha2* is essential for meiotic double-strand break repair in spermatogenesis

Chen Pan[1,2], Keisuke Shimada[1,*,‡], Hsin-Yi Chang[1,3], Haoting Wang[1,2] and Masahito Ikawa[1,2,3,4,5,‡]

## ABSTRACT

It is known that various testis-specific mitochondrial proteins are associated with energy metabolism and male meiosis. PDHA2 is a testis-specific mitochondrial protein, and its encoding gene is speculated to be an autosomal retrogene of the progenitor X-linked *Pdha1*. Here, we show that *Pdha2* knockout (KO) mice exhibit azoospermia due to failure at the late pachytene-diplotene transition. We found that PDHA2 interacts with PDHB and PDHA1. PDHA2 absence leads to decreased PDHB amounts and ATP levels in male germ cells. ATP reduction impairs the function of the ATPase recombination proteins RAD51 and DMC1, causing crossover formation deficiency, further resulting in double-strand break repair failure at the pachytene stage. *Pdha1* expression by transgenes in *Pdha2* KO germ cells rescues fertility and PDHB expression in *Pdha2* KO males, confirming the functional equivalence of PDHA1 and PDHA2. Because X-linked *Pdha1* expression is silenced during meiotic sex chromosome inactivation, our findings also support the hypothesis that *Pdha2* was transposed from *Pdha1*. In summary, PDHA2 compensates for silenced PDHA1 in male germ cells, and plays a crucial role in maintaining efficient double-strand break repair for proper meiotic progression.

KEY WORDS: Meiosis, Double-strand break repair, Meiotic sex chromosome inactivation, Mitochondria, Azoospermia, Mouse

## INTRODUCTION

Retrogenes originate from their progenitor genes by retroposition (Wang, 2004). Several reports have revealed that various retrogenes on the autosomes are derived from X-linked progenitor genes (Uechi et al., 2002). Such a gene often evolves a testis-enriched expression pattern, playing a crucial role in spermatogenesis (Hendriksen et al., 1995). For example, *Rpl10l*, *Utp14b*, *Pgk2*, *Cetn1* and *Hnrnph1* are believed to be retroposed from the X chromosome (Wang, 2004; Korff et al., 2023). Deletion of these retrogenes causes varying

degrees of compromised spermatogenesis (McCarrey and Thomas, 1987; Hart et al., 2001; Rohozinski and Bishop, 2004; Jiang et al., 2017; Korff et al., 2023). Several reports indicate that some X-linked genes are expressed in premeiotic germ cells but not in meiotic and haploid cells due to meiotic sex chromosome inactivation (MSCI) (da Cruz et al., 2016). In contrast, their autosomal retrogene expression increases significantly following MSCI. Therefore, these retrogenes are thought to be transposed from the X chromosome to compensate for the silencing of X-linked progenitor genes by MSCI during the pachytene stage (Sosa et al., 2015). In fact, a previous study has shown that deficiency of the retrogene *Rpl10l*, disruption of which causes meiotic arrest, can be rescued (or compensated for) by expressing its X-linked progenitor gene *Rpl10* by inducing transgenes in germ cells (Jiang et al., 2017). In another study, it was demonstrated that loss of the retrogene *Utp14b* is compensated for by expression of its X-linked progenitor gene *Utp14a* in germ cells (Rohozinski and Bishop, 2004). However, there are still many testis-enriched retrogenes derived from X-linked progenitor genes for which the function is unknown.

It is known that various testis-specific mitochondrial proteins associated with energy metabolism, including glycerol metabolism (GK2), respiratory chain components (PARL) and ATP transporters (SLC25A31, also known as ANT4), are important for spermatogenesis (Brower et al., 2007; Shimada et al., 2019; Schumacher et al., 2024). Comparing the phenotypes of knockout (KO) mice for these testis-specific mitochondrial genes reveals differences in the timing of when the phenotype emerges during spermatogenesis. This variation appears to depend on protein expression timing during spermatogenesis. In addition, these results suggest that mitochondrial proteins affect spermatogenesis by affecting mitochondrial function (Brower et al., 2007; Shimada et al., 2019; Schumacher et al., 2024). MFN1 and MFN2 are testis-specific mitochondrial fusion proteins. KO mice for each of these proteins form spermatozoa with abnormal morphology, but double KO mice show a more severe phenotype with no spermatozoa due to meiotic failure (Varuzhanyan et al., 2019). This study also suggests that meiotic cells require more mitochondria and energy than other germ cells. This highlights the high metabolic requirements of meiotic cells, suggesting that energy deficits during this phase crucially disrupt spermatogenesis.

Mutations of *PDHA2* (pyruvate dehydrogenase E1 subunit alpha 2), encoding a catalytic subunit of the pyruvate dehydrogenase complex (PDC), have been reported to be associated with male infertility in humans (Yildirim et al., 2018; Kherraf et al., 2022). *Pdha2* was speculated to be a retrogene of the X-linked *Pdha1* (pyruvate dehydrogenase E1 subunit alpha 1), based on the properties of their DNA sequences and RNA expression patterns in germ cells (Dahl et al., 1990). In somatic cells, PDHA1 has been shown to form homodimers, which interact with PDHB (pyruvate dehydrogenase E1 subunit beta) to form a heterotetrameric E1 enzyme (Gokcan et al., 2022). Multiple copies of this E1 subunit are assembled into the full PDC complex, which catalyzes the conversion of pyruvate to acetyl-CoA. This process links glycolysis to the tricarboxylic acid

[1]Department of Experimental Genome Research, Research Institute for Microbial Diseases, The University of Osaka, Osaka 5650871, Japan. [2]Department of Experimental Genome Research, Graduate School of Pharmaceutical Sciences, The University of Osaka, Osaka 5650871, Japan. [3]Department of Experimental Genome Research, Graduate School of Medicine, The University of Osaka, Osaka 5650871, Japan. [4]Regulation of Host Defense Team, Center for Infectious Disease Education and Research, The University of Osaka, Osaka 5650871, Japan. [5]Laboratory of Reproductive Systems Biology, The Institute of Medical Science, The University of Tokyo, Tokyo 1088639, Japan.
*Present address: Laboratory of Disease Models, School of Veterinary Medicine, Rakuno Gakuen University, Hokkaido 0698501, Japan.

‡Authors for correspondence (shimada@rakuno.ac.jp; ikawa@biken.osaka-u.ac.jp)

K.S., 0000-0003-3739-7163; M.I., 0000-0001-9859-6217

cycle and supports mitochondrial metabolism (Dabrowska, 1985). A previous study generated *Pdha2* KO mice, and suggested that PDHA2 disruption causes meiotic arrest, possibly due to impaired pyruvate metabolism (Fang et al., 2021). Another study showed that PDHA2 localizes to both the nucleus and cytoplasm of germ cells, implying a potential RNA-binding role in the meiotic nucleus (Li et al., 2024). However, no detailed analysis of meiotic arrest has been done, which means the precise role of PDHA2 remains unclear.

Here, we generated *Pdha2* KO mice using the CRISPR/Cas9 system and revealed that *Pdha2* KO spermatocytes stagnate at the late pachytene stage. *Pdha2* KO germ cells show ATP depletion, double-strand break (DSB) repair failure, and apoptosis. We identified interacting proteins for PDHA2, indicating its involvement in mitochondrial ATP production. Additionally, we revealed experimentally that the X-linked progenitor gene *Pdha1* can compensate for its retrogene *Pdha2* in germ cells. This result supports the hypothesis that *Pdha2* is transposed from the X chromosome to compensate for the silencing of *Pdha1* due to MSCI.

## RESULTS
### PDHA2 localizes to mitochondria and the nucleus in male germ cells

Unlike mouse *Pdha1*, which is localized on chromosome X, the mouse *Pdha2* gene is located on chromosome 3. *Pdha2* has a single

exon, with the transcript encoding a 391 amino acid (aa) protein showing specific expression in mouse testis (Takakubo and Dahl, 1992). PDHA2 is a conserved protein expressed in multiple vertebrate species, including humans and mice, with multiple phosphorylation sites, and is predicted to be a co-factor of pyruvate dehydrogenases binding the thiamine pyrophosphate (TPP) motif (Yildirim et al., 2018). Our data show a 92% similarity between human and mouse PDHA2 (Fig. S1A). We first confirmed that *Pdha2* shows a testis-specific expression by performing reverse transcription polymerase chain reaction (RT-PCR) using multiple tissues from adult mice (Fig. 1A). We then performed RT-PCR using postnatal testis to examine expression timing in germ cells. *Pdha2* expression begins around postnatal day (PND) 6 and becomes strong from PND 12 (Fig. 1B), corresponding to the onset of spermatogonia differentiation and mid-pachytene stage, respectively (Drumond et al., 2011).

To find out the transient localization of PDHA2, we constructed a plasmid vector that contained *Pdha2* with a PA and 1D4 tag under a CMV early enhancer/chicken β-actin (CAG) promoter (Fig. S1B). We transiently expressed the plasmid in COS-7 cells and observed PDHA2 localization. PDHA2 colocalized with the mitochondrial marker TOM20 (TOMM20) (Fig. S1C). To reveal PDHA2 localization *in vivo*, we performed immunofluorescence (IF) analysis using testis sections. Anti-COX IV (a mitochondrial marker)

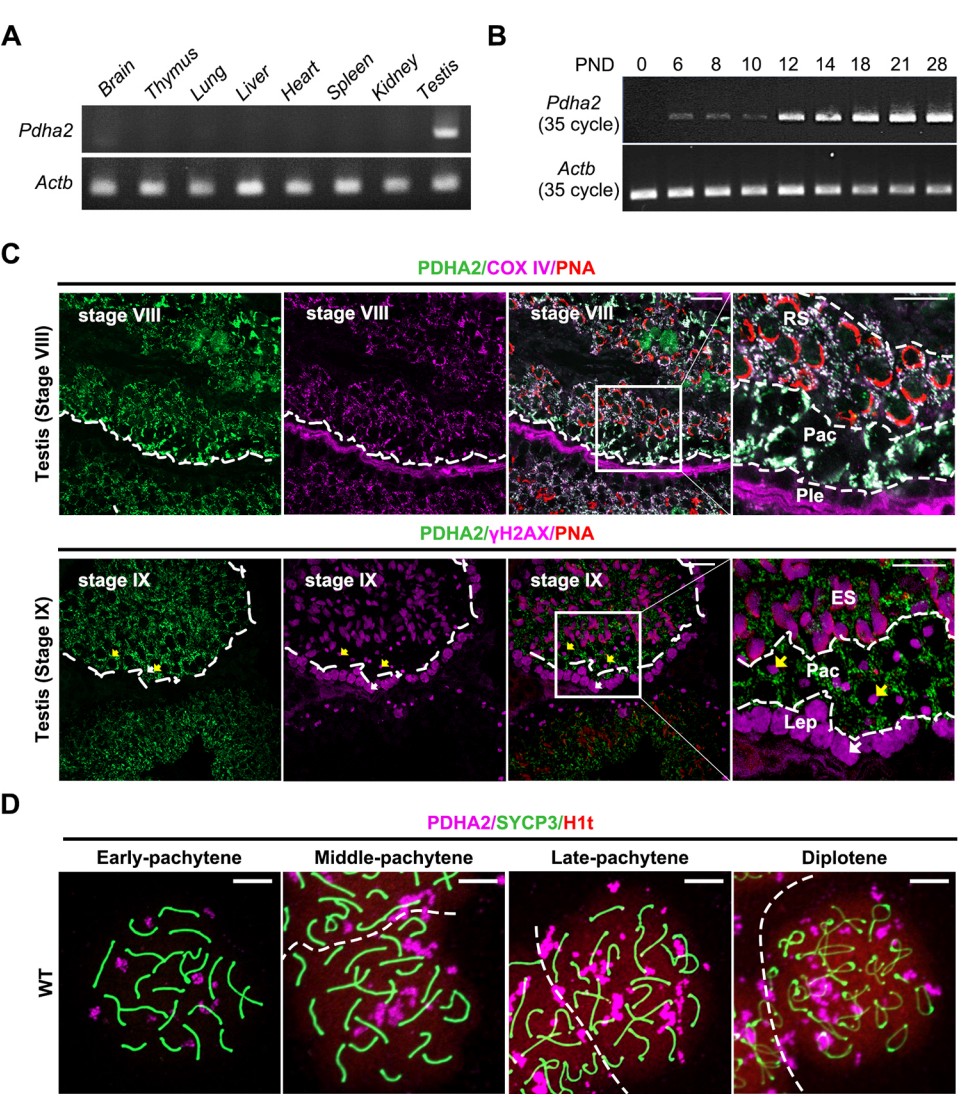

**Fig. 1. PDHA2 localizes in both mitochondria and the nucleus in germ cells.** (A,B) RT-PCR of *Pdha2* using RNA from various mouse tissues (A) and postnatal testes (B). *Actb* was used as a loading control. (C) Co-immunofluorescence analysis using WT adult testis. Spermatogenic stages were distinguished by the morphology of the acrosome and nucleus stained with lectin PNA (red) and Hoechst 33342 (not shown here), respectively. PDHA2 (green) is colocalized with COX IV (mitochondrial marker; magenta) from the pachytene stage of meiosis (top panels) to spermatids, and not expressed at the leptotene stage (bottom panels). The γH2AX signal (magenta) indicates DNA damage, which is located throughout the nucleus before the pachytene stage, and appears on the sex chromosome as one body (yellow arrows) during the pachytene and diplotene stages. Scale bars: 20 µm. The white dashed lines indicate cells with different layers during spermatogenesis. ES, elongating spermatids; Lep, leptotene spermatocytes; Pac, pachytene spermatocytes; Ple, preleptotene spermatocytes; RS, round spermatids. Images are representative of three biologically independent mice with consistent results. (D) IF analysis of spread nuclei from prophase I spermatocytes collected from PND 21 WT mice. Prophase I spermatocytes were immunostained with SYCP3 (green), PDHA2 (magenta) and H1t (red). H1t is a marker for the pachytene stage. H1t first appears in the mid-pachytene spermatocytes, and expression increases significantly in the later stages. Dashed lines indicate the boundary between two adjacent nuclei. Scale bars: 10 µm. Image are representative of three independent mice showing consistent findings.

or anti-γH2AX (a meiotic DNA damage marker) antibodies were used for co-staining. IF analysis determined that PDHA2 localized in mitochondria from the pachytene stage of mouse spermatogenesis (Fig. 1C, Fig. S2A). Because PDHA2 has been reported to be expressed in the nuclear fraction (Li et al., 2024), we next performed chromosome spreading to clarify the localization expression pattern in the nucleus. We confirmed the clustered-like granule localization of PDHA2 within the nucleus by co-staining with SYCP3 (a marker of meiotic chromosome axes) and H1t (H1f6; detected in mid-pachytene and increased in intensity in late-pachytene and diplotene spermatocytes) on smeared germ cells collected from PND 21 wild-type (WT) testes (Fig. 1D). Therefore, we speculate that PDHA2 may play a role in the nucleus as well as the mitochondria during male meiosis.

## PDHA2 is essential for meiosis completion and male fertility

To investigate the function of PDHA2 in male fertility, we generated *Pdha2* KO (*Pdha2*$^{-/-}$) mice using the CRISPR/Cas9 system. PCR using genomic DNA and Sanger sequence revealed a *Pdha2* mutant line having a 2374 base pair deletion, which disrupts almost all open reading frames (Fig. 2A,B). We confirmed the loss of PDHA2 protein by western blot (WB) analysis and IF analysis in adult KO testes (Fig. 2C, Fig. S2B).

To assess fertility, *Pdha2*$^{-/-}$ and WT males were individually housed with three WT females for 2 months. None of the *Pdha2*$^{-/-}$ males produced pups, indicating that *Pdha2*-deleted male mice are sterile (Fig. 2D). *Pdha2*$^{+/-}$ males produced offspring in breeding cages and were considered fertile (Fig. S3A); thus, heterozygous male littermates were used as controls, generated by mating *Pdha2*$^{+/-}$ males

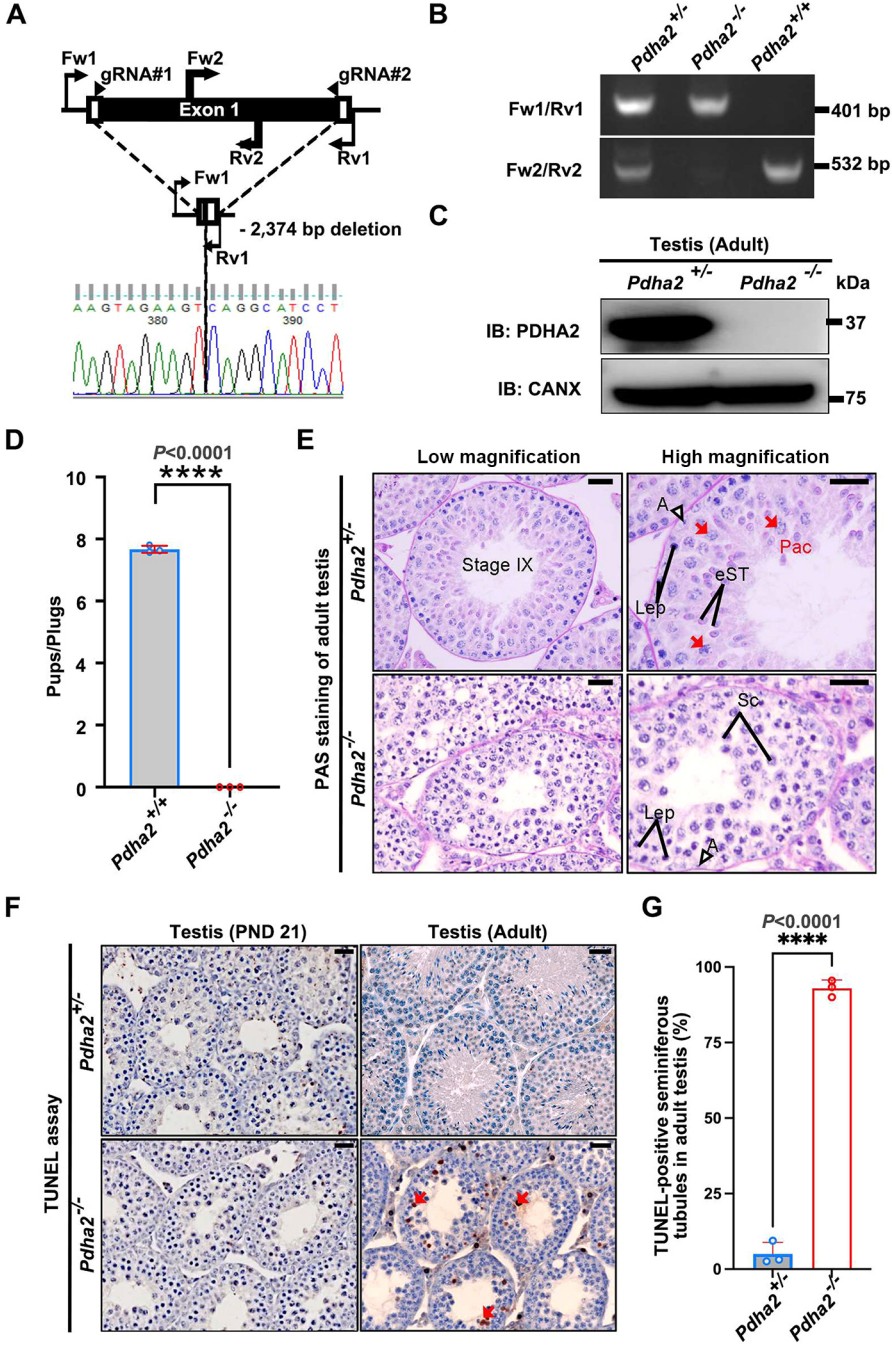

Fig. 2. *Pdha2* KO male mice are infertile with abnormal spermatogenesis. (A) Diagram for generating *Pdha2* KO mice using the CRISPR/Cas9 system. White boxes represent untranslated regions, black boxes indicate protein-coding regions. The two gRNAs are shown with polarity indicated. Fw1/2 and Rv1/2 denote the forward and reverse primers used for genotyping, respectively. (B) Genotyping of the obtained *Pdha2*-deletion mice was performed using Fw#1-Rv#1 primers for the KO allele, and Fw#2-Rv#2 primers for the WT allele. (C) Protein expression of PDHA2 in *Pdha2*$^{+/-}$ and *Pdha2*$^{-/-}$ mice testes. CANX was used as a loading control. IB, immunoblot. (D) Number of litters born per plug. Individual adult WT and *Pdha2*$^{-/-}$ male mice (*n*=3) were mated to three WT female mice. Data shown as mean±s.d.; two-tailed, unpaired *t*-test. (E) Histology of adult testes at different magnifications. In stage IX seminiferous tubules, multiple layers of cells are observed in *Pdha2*$^{+/-}$ mice, including spermatocytes and spermatids. *Pdha2*$^{-/-}$ mice only show spermatocytes, with no round spermatids present. Red arrows indicate pachytene spermatocytes. A, undifferentiated type A spermatogonia; eST, elongated spermatids; Lep, leptotene spermatocytes; Pac, pachytene spermatocytes; Sc, spermatocytes. Scale bars: 40 µm. (F) TUNEL staining of seminiferous tubules in PND 21 and adult mice counterstained with Hematoxylin. No differences were observed in PND 21 testis, but apoptotic spermatocytes near the lumen were observed in adult *Pdha2*$^{-/-}$ seminiferous tubules compared to control. Red arrows indicate TUNEL-positive cells. Scale bars: 40 µm. (G) Quantitative analysis of the percentage of tubules containing TUNEL-positive cells in control and *Pdha2* KO adult mice (*n*=3). Data shown as mean±s.d.; two-tailed, unpaired *t*-test. ****P*<0.0001.

with *Pdha2⁻/⁻* females. We next measured the testis weight of 12-week-old mice, and found that *Pdha2⁻/⁻* testes are half the weight of *Pdha2⁺/⁻* testes (Fig. S3B-D). These results suggest that male infertility may be caused by abnormal spermatogenesis. We then examined testis sections and found the absence of metaphase II spermatocytes and spermatids in KO adult testes (Fig. 2E). In KO cauda epididymis, abnormal round cells were observed instead of spermatozoa (Fig. S3E). A terminal deoxynucleotidyl transferase (TdT)-mediated dUTP-biotin nick end labeling (TUNEL) assay revealed a higher proportion of apoptotic spermatocytes in adult mutant mice than in controls, but no differences were observed in PND 21 testis sections (Fig. 2F,G). The results of these experiments indicate that PDHA2 absence results in germ cell apoptosis, abnormal meiosis completion and azoospermia, leading to male infertility.

### PDHA2 is required for the late pachytene-diplotene transition in male meiosis

To track the specific stage at which spermatocytes are arrested in *Pdha2⁻/⁻* males, we analyzed meiosis progression by IF analysis using PND 21 germ cells. We performed chromosome spreading, and IF analysis using antibodies against SYCP3 and γH2AX (a marker of DNA DSBs for meiotic recombination and XY bodies). IF results revealed that no diplotene spermatocytes were observed in *Pdha2* KO testis, indicating pachytene arrest during meiosis (Fig. 3A,B). To investigate further which stage of pachytene spermatocytes exhibit the defect in *Pdha2⁻/⁻* mice, we chose an antibody against histone H1t (a marker of the mid- to late-pachytene stage) for IF analysis. The IF results demonstrated remarkable stagnation in late-pachytene spermatocytes in *Pdha2⁻/⁻* mice (Fig. 3C,D).

### *Pdha2* KO mice exhibit normal MSCI and autosome synapsis

MSCI, the first checkpoint during pachytene spermatocytes, is a pivotal process in male meiosis that ensures the proper progression of spermatogenesis (Fig. 3E) (Turner, 2015). The X and Y chromosomes are largely non-homologous and do not undergo chromosome pairing (Kauppi et al., 2011). Proper synapsis (pairing of homologous chromosomes) of autosomes is crucial for MSCI execution, and failure in synapsis will result in meiotic arrest (Sanchez-Saez et al., 2020). To check whether synapsis formation occurred during meiosis, we used SYCP1 as a synapsis marker. IF analysis revealed that SYCP1 is expressed on autosomes except the sex chromosomes in both control and KO meiotic cells (Fig. S4A). Because MSCI is initiated by BRCA1 localization in early-pachytene spermatocytes (Abe et al., 2022), we investigated BRCA1 localization in *Pdha2⁻/⁻* pachytene spermatocytes and found no changes (Fig. 3F). Ataxia telangiectasia mutated and Rad3-related (ATR) kinase is a key regulator of the DNA damage response (DDR) (Abe et al., 2022). ATR is recruited by BRCA1, which phosphorylates H2AX, converting it to γH2AX (Krum et al., 2010). γH2AX amplification and expansion of signals from the pseudoautosomal region (PAR) in mid-pachytene spermatocytes form XY bodies, which indicates MSCI completion (Fig. 3E). IF analysis revealed γH2AX expression on the XY body in KO meiotic cells, which indicates correct localization (Fig. 3A). Furthermore, RNA polymerase II (Pol II) is associated with active transcription (Fenstermaker et al., 2023). During MSCI, the sex chromosomes are transcriptionally silenced, and Pol II is inactivated in the XY body (Alavattam et al., 2021; Alexander et al., 2023). We observed Pol II localization in meiotic cells, and found no differences between control and KO spermatocytes (Fig. 3G). We next explored H3K9 trimethylation (H3K9me3) localization during the pachytene stage. H3K9me3 is required for maintaining MSCI by recruitment of silencing factors and

coordination with DDR, leading to stable and irreversible MSCI establishment on X pericentric chromatin (X-PCH) and a part of the Y chromosome at mid-pachytene stage, which then disappears from the Y chromosome at late-pachytene stage (Abe et al., 2022; Alavattam et al., 2024). H3K9me3 showed normal localization in both *Pdha2⁺/⁻* and *Pdha2⁻/⁻* meiotic cells (Fig. S4B). Taken together, our results demonstrate that *Pdha2* deletion does not affect autosomal synapsis and MSCI in meiotic cells despite an abnormal pachytene-diplotene transition.

### PDHA2 mutant spermatocytes exhibit decreased crossover due to compromised DSB repair

A second checkpoint in pachytene spermatocytes, DSB repair, is predominantly mediated by homologous recombination (HR), a high-fidelity mechanism that utilizes the homologous chromosome as a template (Ruiz-Herrera and Waters, 2022). Many key proteins that contribute to DSB repair process have been identified (Dapper and Payseur, 2019), including RPA2, RAD51, DMC1, MSH4 and MLH1 (Fig. 4A). Sufficient MLH1 signals at crossover sites on chromosomal axes indicate successful DSB repair, and are crucial for the pachytene-diplotene transition (Edelmann et al., 1996). MLH1 foci on the chromosomal axis were therefore counted. IF analysis revealed dramatically decreased MLH1 foci in KO pachytene spermatocytes compared to controls (Fig. 4B,C), suggesting DSB repair failure. To clarify the precise steps at which DSB repair processes became abnormal, we conducted IF analysis sequentially using antibodies targeting the proteins involved in the DSB repair processes. HORMAD1 is a marker indicative of DSB occurrence (Daniel et al., 2011). Compared to controls, there were no differences in HORMAD1 localization on X and Y chromosomes during pachytene stages, indicating that DSBs occur normally in KO meiotic cells (Fig. S4C). Next, we explored the expression pattern of RPA2, which binds to single-stranded DNA to prevent secondary structure formation (Shi et al., 2019). Our data demonstrated normal RPA2 signals and amounts in *Pdha2* KO pachytene spermatocytes (Fig. S5A-D). RPA2 stabilizes the single-stranded DNA and facilitates the recruitment of other proteins involved in HR during the DSB repair process, such as RAD51 (Anand et al., 2022). We therefore performed IF analysis using an anti-RAD51 antibody. Our data suggested a dramatic decrease in the number of RAD51 foci in KO mice at mid- to late-pachytene stages compared to controls, but not at the early-pachytene stage (Fig. 4D,E, Fig. S5A,D). DMC1 is a meiotic-specific recombination protein that interacts with RAD51 during DSB repair (Ito et al., 2023). When we examined DMC1 localization, a remarkable reduction of DMC1 foci in *Pdha2* KO mice from mid-pachytene stages was observed (Fig. 4D,F, Fig. S5A,D). MSH4 forms a heterodimer (known as MutSγ) with MSH5 to stabilize recombination intermediates, such as double Holliday junctions and single-end invasions, strengthening DSBs repair (Novak et al., 2001). We observed decreased numbers of MSH4 foci on the chromosome axis in *Pdha2* KO mid/late-pachytene spermatocytes (Fig. 4D,G, Fig. S5A).

We also observed segregation of X and Y chromosomes in *Pdha2* KO late-pachytene spermatocytes (Figs 3F,G and 4D, Fig. S4B,C). When we examined sex chromosomes in late-pachytene spermatocytes, the majority of KO spermatocytes exhibited abnormal separation of X and Y chromosomes (Fig. 4H). Consistent with our results, X and Y chromosome premature separation has been reported in *Rad51ap2*-mutated pachytene spermatocytes, which is caused by impaired DSB repair

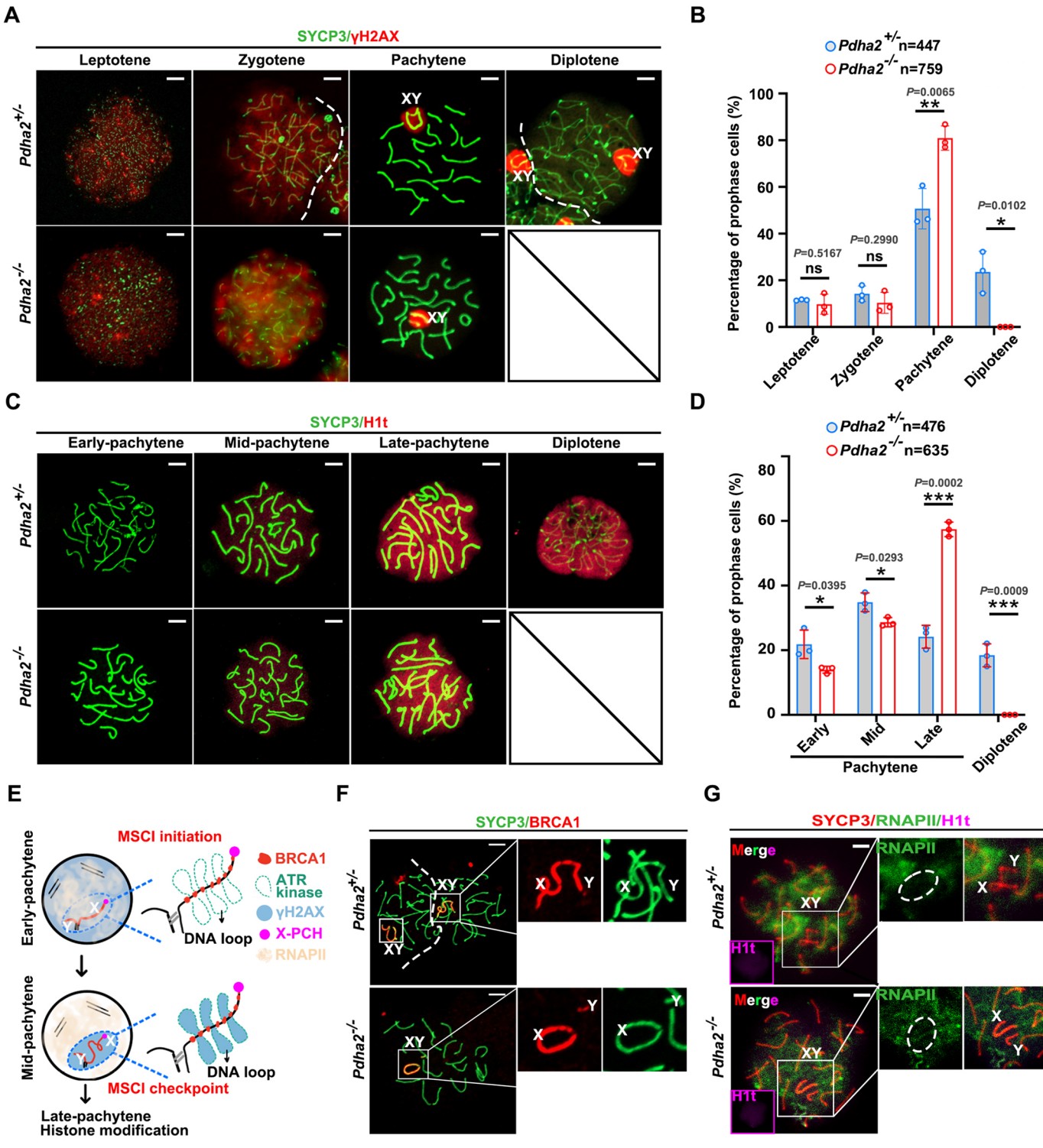

**Fig. 3.** See next page for legend.

in the PAR due to recombination instability (Ma et al., 2022). However, these mice do not exhibit pachytene stagnation during male meiosis. This means that X and Y chromosome separation is independent of the abnormal pachytene-diplotene transition. Overall, our findings indicate that abnormal HR occurs during mid- and late-pachytene stages in *Pdha2⁻/⁻* mice, causing DSB repair failure and preventing spermatocytes from developing into diplotene stages.

## PDHA2 interacts with PDHB and PDHA1 in the testis

To analyze the role of PDHA2 at the molecular level, we identified PDHA2-interacting proteins by immunoprecipitation (IP) using testes obtained from 10-week-old male mice. SDS-PAGE and silver staining after IP confirmed several specific bands in the experimental samples as expected (Fig. 5A). Mass spectrometry (MS) analysis using the immunoprecipitant revealed several interactomes (Fig. 5B). The 3′-UTR of PDHX harbors a potential

**Fig. 3. *Pdha2* KO spermatocytes fail to transition from the late-pachytene stage to diplotene stage.** (A) Spread nuclei of prophase spermatocytes collected from *Pdha2*$^{+/-}$ and *Pdha2*$^{-/-}$ male mice were stained with anti-SYCP3 (green) and γH2AX (red) antibodies. XY indicates the sex chromosomes encircled by γH2AX signal. Dashed lines indicate the boundary between adjacent nuclei. At least three male mice were analyzed. Scale bars: 5 µm. Experiments were repeated at least twice with consistent results. (B) Distribution of different meiotic prophase I stages in *Pdha2*$^{+/-}$ (gray bars) and *Pdha2*$^{-/-}$ (red open bars). Data shown as mean±s.d. Statistical analysis by two-tailed, unpaired *t*-test; ns, no significance. (C) Spread nuclei of prophase spermatocytes collected from *Pdha2*$^{+/-}$ and *Pdha2*$^{-/-}$ male mice were immunostained with anti-SYCP3 (green) and H1t (red) antibodies. Fluorescence images were pseudocolored using ImageJ. At least three male mice were analyzed. Scale bars: 5 µm. Experiments were repeated at least twice with consistent results. (D) Distribution of different pachytene and diplotene stage in *Pdha2*$^{+/-}$ (gray bars) and *Pdha2*$^{-/-}$ (red open bars). Data shown as mean±s.d. Statistical analysis by two-tailed, unpaired *t*-test. (E) Model of MSCI (meiotic sex chromosome inactivation) checkpoint during pachytene stage. BRCA1 as one of the earliest DDR factors binds to the unsynapsed axes (XY), and recruits ATR to phosphorylate H2AX, converting it to γH2AX. γH2AX signals DNA damage and recruits other DDR proteins to propagate the silencing signal across the sex chromosomes. Simultaneously, the XY body will form, and the completion of MSCI leads to the cessation of RNA transcription on sex chromosomes at the mid-pachytene stage (Abe et al., 2020). Subsequently, various histone modifications occur to regulate gene expression and chromatin structure. (F) Representative nuclear spreads stained for SYCP3 (green) and BRCA1 (red) in pachytene-stage PND 21 mice. Magnified views of the X and Y chromosomes are shown on the right. Dashed lines indicate the boundary between adjacent nuclei. Scale bars: 5 µm. Experiments were repeated at least twice with consistent results. (G) Representative nuclear spreads stained for SYCP3 (red), RNAPII (red) and H1t (magenta) during the mid-pachytene stage of PND 21 mice. The X and Y chromosomes are outlined in white. Enlarged images of the XY body region are displayed on the right. Widefield images of H1t staining (magenta) are shown in the left lower corner of the overlay images. Scale bars: 5 µm. Experiments were repeated at least twice with consistent results. *$P$<0.05, **$P$<0.001, ***$P$<0.001.

target site for miRNA-26a, which may directly target DNA methyltransferases or histone deacetylases (Ford, 1970; Wang et al., 2020). TARDBP is a splicing-related protein that is transcriptionally upregulated in BRDT-deficient spermatids, acting as a downstream effector in BRDT-mediated mRNA processing, which is essential for proper post-transcriptional regulation during spermiogenesis (Berkovits et al., 2012; Manterola et al., 2018). HDAC6 impairs DNA damage repair in a deacetylase activity-dependent manner by modulating histone modifications (Qiu et al., 2023). These proteins suggest PDHA2 may be an epigenetic regulator during meiosis. To note, most of the identified proteins, e.g. PDHB, PDHX, DLD, PDHA1 and PDK3 (Fig. S6A), are related to the PDC and mitochondrial metabolism (Imbard et al., 2011). Gene ontology (GO) analysis suggests that the most downregulated pathway is related to the PDC, the acetyl-coA process and the tricarboxylic acid cycle (Fig. S6B). A comparison of the expression patterns of these proteins during spermatogenesis revealed that PDHB exhibited the most similar patterns to PDHA2, as well as the highest quantitative value in MS data (Fig. 5B, Fig. S6C). To confirm the interactions between PDHA2, PDHA1 and PDHB, we performed IP and subsequent WB analyses, which demonstrate that PDHA2 interacts with both PDHB and PDHA1 (Fig. 5C, Fig. S6D).

Furthermore, alignment of mouse PDHA2 and PDHA1 amino acid sequences indicated 76% identity and 95% similarity (Fig. S7A). *Pdha1* is a ubiquitously expressed gene, whereas *Pdha2* is a testis-specific gene (Fig. 1A, Fig. S7B). Because of this high similarity, we carefully validated the specificity of anti-PDHA1 and anti-PDHA2 antibodies to exclude potential cross-reactivity. We performed WB analysis using cell lysates after overexpressing epitope-tagged *Pdha1*

and *Pdha2* (Figs S1B and S7C) in HEK293T cells. Both antibodies specifically recognized their target proteins as expected (Fig. S7D). We further verified antibody specificity using liver tissue as a negative control. IP-WB analysis showed no PDHA2 signal in liver lysates and no PDHA1 signal in liver immunoprecipitants using an anti-PDHA2 antibody (Fig. S7E). These results collectively confirm that there is no cross-reactivity between anti-PDHA1 and anti-PDHA2. Given that *Pdha2* is a predicted retrogene gene derived from X-linked *Pdha1* (Dahl et al., 1990), we propose that the composition of the E1 component of the PDC undergoes a dynamic shift during spermatogenesis. Although the canonical E1 tetramer in somatic cells consists of two PDHA1 and two PDHB subunits (Gokcan et al., 2022), we speculate that, in male germ cells, PDHA2 may gradually replace PDHA1 to form a testis-specific E1 configuration. This transition likely proceeds through an intermediate form that contains PDHA1, PDHA2 and PDHB (Fig. 5C, Figs S6D and S7E), eventually yielding a complex composed of two PDHA2 and two PDHB subunits. Although PDHA2 may not directly interact with PDHA1, its strong association with PDHB suggests that it is a functional component of the testis-specific PDC, essential for mitochondrial metabolism in germ cells.

## The absence of PDHA2 and its interactomes in testicular mitochondria lead to reduced ATP amounts in germ cells

We then explored the localization of PDHA2-interacting protein PDHB in the juvenile testis (PND 21). IF analysis suggested that PDHB localized to mitochondria in spermatogonia and spermatocytes (Fig. 5D). The fluorescence intensity and protein amounts of PDHB in juveniles were dramatically decreased in *Pdha2* KO mice compared to controls (Fig. 5D, Fig. S8A). This may be due to the instability of the PDC due to the absence of PDHA2. As a note, other mitochondrial proteins showed no change in protein amounts (Fig. S8A). We did not observe any nuclear localization of PDHB in pachytene spermatocytes (Fig. S8B). In contrast to PDHB, PDHA1 in the juvenile testis was localized to both mitochondria and the nucleus in spermatocytes (Fig. 5E). These PDHA1 signals in the nucleus were colocalized with the XY body during meiosis (Fig. 5F). To understand more about the nuclear localization of PDHA1 during pachytene spermatocytes, we conducted IF after chromosome spreading. Two expression patterns of PDHA1 were revealed in the nucleus (Fig. S8C): cluster-like granules similar to PDHA2 localization (blue arrowheads), and PAR regions on XY bodies (white arrowheads). Although PDHA1 fluorescence intensity in mitochondria and the nucleus decreased in *Pdha2* KO spermatocytes compared to controls (Fig. 5E,F), PDHA1 was still present on the X and Y chromosomes in KO spermatocytes (Fig. S8C), and the expression of cluster-like granules was detected irregularly in both control and KO nuclei (Fig. S8C). Thus, PDHA1 was still detected in small amounts after MSCI due to its persistent nuclear localization.

PDHA2, PDHB and PDHA1 were all localized in the mitochondria in spermatocytes. To examine whether these proteins are simply attached to the mitochondrial membrane or are transmembrane proteins, we performed topological analysis using mitochondrial fractions purified from the testis. Ultrasonic treatment of the mitochondrial fraction followed by exposure to $Na_2CO_3$ to increase the pH results in disruption of mitochondrial structure. Alkaline conditions can compromise the integrity of the membrane structure, especially weak electrostatic and hydrophobic interactions with the membrane. Peripheral (proteins attached to the membrane) mitochondrial proteins (such as ATP5A) are released into the soluble fraction. In contrast, integral membrane proteins (such as VDAC1) remain embedded in the membrane because of their

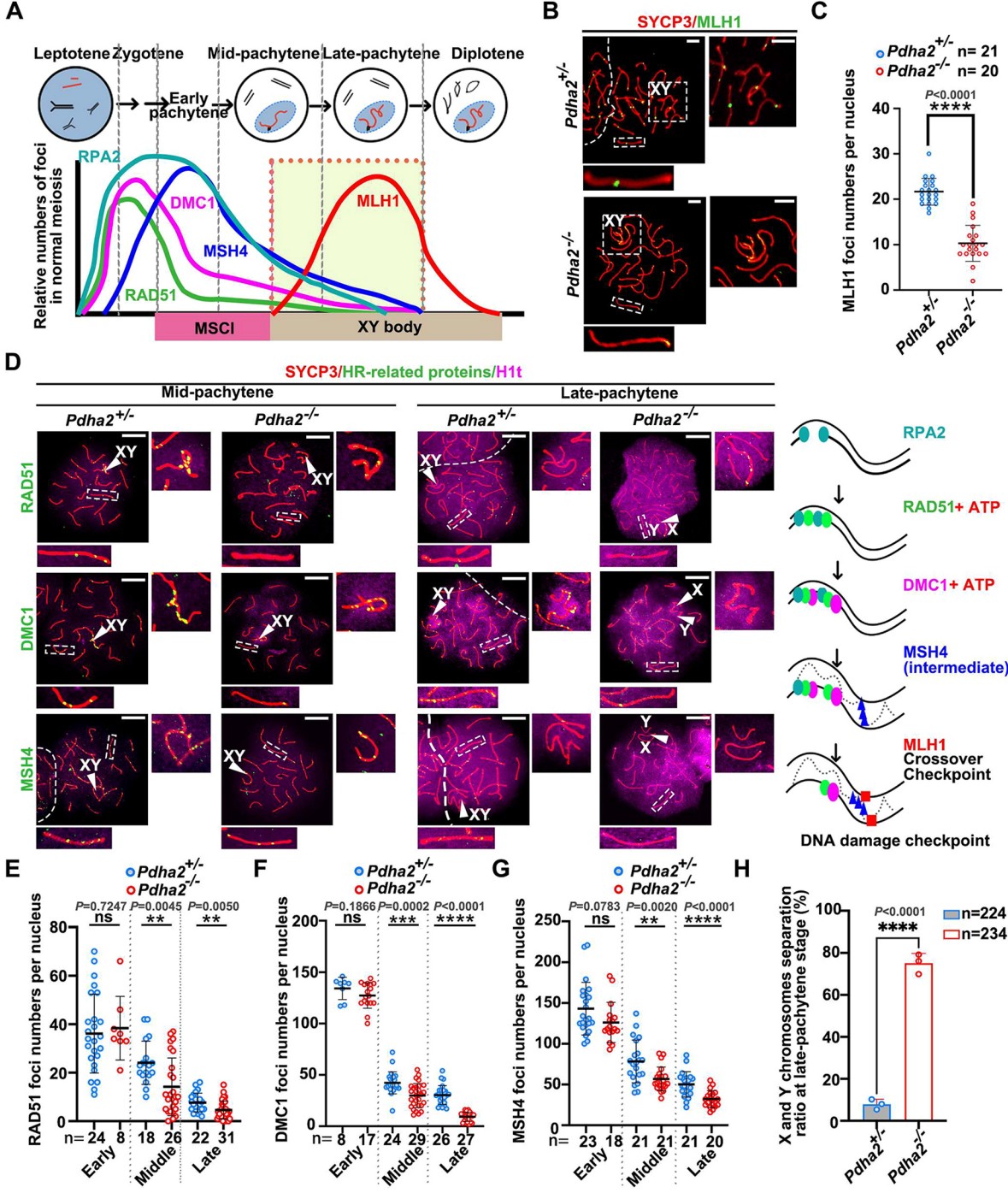

**Fig. 4. Decreased foci on the chromosome axis of meiotic recombination proteins in *Pdha2* KO spermatocytes.** (A) Model of the two pachytene checkpoints and their relationship to meiotic recombination. MSCI is the first checkpoint at the mid-pachytene stage, marked by XY body formation. RPA2, RAD51, DMC1, MSH4 and MLH1 are the central recombination proteins for different steps during DSB formation and repair, including homology search and strand invasion. The relative number of foci on the chromosomes is shown. (B) Representative spread nuclei of spermatocytes stained for SYCP3 (red) and MLH1 (green) at the pachytene stage using PND 21 mice. PAR and an autosome are indicated inside the dashed white boxes, and magnified views are shown on the right. Dashed lines indicate the boundary between adjacent nuclei. Scale bars: 5 µm. (C) Frequencies of nuclei with an MLH1 focus detected on chromosome axes in *Pdha2*[+/−] and *Pdha2*[−/−] pachytene spermatocytes from three PND 21 mice. Data shown as mean±s.d. (*n*=3 mice). Each point corresponds to the number of foci in a spermatocyte nuceleus. The results of two-tailed, unpaired *t*-tests are indicated in the graphs. (D) Representative spread nuclei of mid- and late-pachytene spermatocytes (marked by H1t; magenta) stained for SYCP3 (red) and DSB repair representative proteins (green) using PND 21 mice (see A). Arrowheads indicate the XY body, and magnified views of sex chromosomes are shown on the right. An autosome is indicated inside the dashed white boxes, and a magnified view is shown at the bottom. Dashed lines indicate the boundary between adjacent nuclei. Scale bars: 10 µm. Schematics of recombinant protein recruitment in order are shown on the right. (E-G) Foci numbers per cell of RPA2/RAD51/DMC1/MSH4 localization on the chromosome axis in the nucleus. *Pdha2*[+/−] and *Pdha2*[−/−] at different pachytene stages of pachytene spermatocytes were examined. Data shown as mean±s.d. (*n*=3 mice). Each point corresponds to the number of foci in a spermatocyte nuceleus. The results of two-tailed, unpaired *t*-tests are indicated in the graphs. ns, no significance. (H) X and Y chromosome separation ratio of late pachytene spermatocytes in PND 21 *Pdha2*[+/−] (gray in blue bars) and *Pdha2*[−/−] (red open bars) mice. Data shown as mean±s.d. (*n*=3 mice). Statistical analysis by two-tailed, unpaired *t*-test. \*\**P*<0.01, \*\*\**P*<0.001, \*\*\*\**P*<0.0001.

transmembrane domains, which can be detected in the pellets after centrifugation. By WB analysis, we showed the same expression pattern of PDHA2, PDHB and PDHA1 as was observed for ATP5A (Fig. 5G), suggesting that they are peripheral mitochondrial proteins. Furthermore, a proteinase K protection assay was performed to detect intracellular/extracellular localization of proteins. Proteinase K digests proteins exposed on the surface of intact organelles but cannot access the domains protected by membranes. Thus, proteinase K treatment

of intact mitochondria without the detergent Triton X-100 leads to the loss of immunoblot signals for outer mitochondrial membrane proteins (such as TOM20) but not inner mitochondrial membrane proteins (such as UQCRC1). Our WB analysis revealed that immunoblot signals for PDHA2, PDHB and PDHA1 were consistent with those of UQCRC1 (Fig. 5H), suggesting that these proteins are inner mitochondrial membrane proteins. Taken together, these results indicate that these three proteins are peripheral

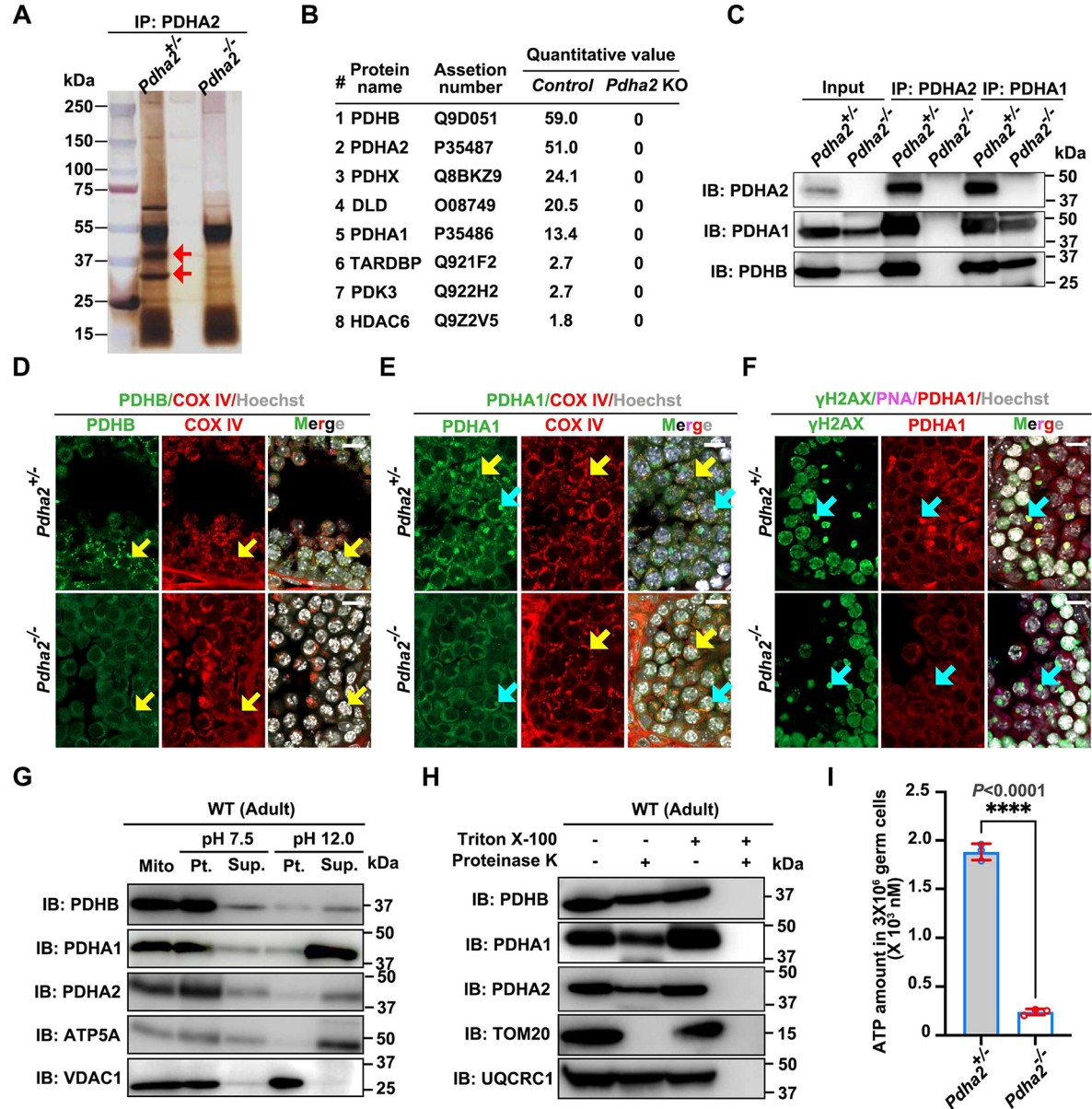

**Fig. 5. PDHA2 absence induces low ATP levels in germ cells.** (A) Silver staining of adult testis lysates after immunoprecipitation (IP) using an anti-PDHA2 antibody. Red arrows indicate specific bands observed in the experimental sample. (B) The list of identified proteins by MS analysis after IP with anti-PDHA2 antibody. The quantitative value was calculated using Scaffold5. (C) WB analysis after IP with anti-PDHA2, anti-PDHA1 and anti-PDHB antibodies in *Pdha2*$^{+/-}$ and *Pdha2*$^{-/-}$ adult testes. IB, immunoblot. (D,E) PDHB (green), PDHA1 (green) and COX IV (red) IF analysis of PND 21 testis. Hoechst 33342 (white) was used for visualizing nuclei. The yellow arrows show PDH proteins colocalized with COX IV, while the blue arrows show PDHA1 localized in the nucleus. Scale bars: 10 µm. (F) PDHA1 (red) and γH2AX (green) IF analysis using PND 21 testis. Lectin PNA (magenta) and Hoechst 33342 (white) were used for visualizing the acrosome and nuclei, respectively. The blue arrows indicate PDHA1 colocalized with the XY body. Scale bars: 10 µm. Images in D-F are representative of two independent samples. (G) The mitochondrial fraction from testis was subjected to ultrasonic treatment, incubated with or without Na$_2$CO$_3$, and then centrifuged to yield a membrane fraction (Pt.) and a soluble fraction (Sup.). These fractions were then subjected to immunoblot analysis with the indicated antibodies. (H) The mitochondrial fraction from adult testis was incubated in the absence or presence of proteinase K and Triton X-100, and subjected to immunoblot analysis with the indicated antibodies. (I) ATP amount in germ cells in PND 21 testis samples. Data are presented as mean ±s.d. The result of a two-tailed, unpaired *t*-test is shown. ****$P<0.0001$.

mitochondrial proteins attached to the inner membrane. As PDHA1, PDHA2 and PDHB are members of the PDC, we can predict that these three proteins have a crucial role in mitochondrial metabolism in the testis. Unsurprisingly, we demonstrated that ATP levels in juvenile germ cells were significantly reduced in *Pdha2* KO mice (Fig. 5I).

### *Pdha1* expression in germ cells restores infertility in *Pdha2* KO males

During MSCI, many X-linked genes are transcriptionally silenced to prevent overexpression and ensure proper meiosis (Loda et al., 2022). To compensate, some genes have given rise to autosomal retrogenes via reverse transcription and genomic integration during evolution (Juchniewicz et al., 2021). These retrogenes are often expressed specifically in the male germline and are thought to substitute functionally for the silenced X-linked genes during spermatogenesis (Wang, 2004). We compared the DNA sequence and coding sequence of *Pdha1* and *Pdha2* (Fig. S9A) and found that *Pdha2* lacks introns and consists of a single exon. The *Pdha2* gene sequence is almost identical to the *Pdha1* coding sequence (Fig. S9A), which suggests that *Pdha2* is of retrotransposon origin (Fig. S9B). X-chromosome-linked *Pdha1* expression declined from meiosis onset (leptotene spermatocytes) to undetectable levels in pachytene spermatocytes (Hermann et al., 2018; Fang et al., 2021), whereas autosome-linked *Pdha2* showed a dramatic increase in the pachytene stage at the mRNA level (Fig. S9C). Therefore, we speculate that PDHA1 and PDHA2 have similar functions, and PDHA2 compensates for the reduction of PDHA1 expression in germ cells, which is due to silencing by MSCI.

To test our hypothesis, we generated transgenic (Tg) mouse lines that express *Pdha1* in germ cells. Before generating Tg mice, we constructed a plasmid vector with PA- and 1D4-tagged *Pdha1* under the CAG promoter (Fig. S7C). This plasmid vector was overexpressed in HEK293T cells, and WB analysis confirmed epitope-tagged PDHA1 expression (Fig. S7D). We then overexpressed the vector in COS-7 cells, and IF analysis revealed epitope-tagged PDHA1 localized to mitochondria (Fig. S9D). We then generated Tg mice expressing PA- and 1D4-tagged *Pdha1* under the germ cell-specific *Clgn* promoter, driving *Pdha1* expression in spermatocytes and spermatids (Ikawa et al., 2001) (Fig. 6A). Presence of the transgene was validated by PCR (Fig. 6B), and WB analysis using an anti-1D4 antibody revealed expression of the transgene in the testis (Fig. 6C). We then mated *Pdha2* KO male mice with the epitope-tagged *Pdha1* transgene (referred to as *Pdha2*$^{-/-}$+Tg) with WT females, and observed pups produced in comparable numbers to control (Fig. 6D). This result demonstrates that *Clgn* promoter-driven *Pdha1* rescues *Pdha2*$^{-/-}$ male fertility.

We then evaluated PDHA1 behavior in Tg mice. WB analysis was performed to determine whether the decreased expression of PDHB in *Pdha2* KO germ cells (Fig. 5D, Fig. S8B) was restored by PDHA1 expression after MSCI using the Tg mice. WB analysis revealed that transgenic PDHA1 expression after MSCI ameliorates PDHB expression (Fig. 6E). In addition, we demonstrated that the PDHA1 transgene rescued PDHB localization to mitochondria in *Pdha2* KO germ cells (Fig. 6F). In the nucleus, we found cluster-like granules of both PDHA1 and PDHA2 that were colocalized (Fig. 6G). Taken together, we conclude that PDHA1 expression after MSCI in germ cells can compensate for PDHA2 function in mitochondria and nuclei during spermatogenesis in *Pdha2* KO male mice.

### DISCUSSION
In this study, we showed that the testis-specific gene *Pdha2* (Fig. 1A) is essential for male fertility (Fig. 2D). Deletion of *Pdha2*

leads to azoospermia due to meiotic arrest at the late pachytene-diplotene stage transition (Figs 2E,F and 3A-D, Fig. S3E). In addition, the study also shows that *Pdha2* KO mice exhibit normal MSCI (Fig. 3A,E-G), and that *Pdha1*, the X-linked homolog, is transcriptionally silenced during MSCI (Figs S7A and S9A,C). We note that PDHA1 protein localization to the XY body is consistent with this silencing (Fig. 5E,F, Figs S6C and S8C). The expression of *Pdha1* in germ cells after MSCI rescues male fertility in *Pdha2* KO mice (Fig. 6D), indicating functional redundancy between PDHA1 and PDHA2 when temporal expression is matched. Our data support the hypothesis that *Pdha2* is a retrogene derived from X-linked *Pdha1* (Dahl et al., 1990), and that *Pdha2* evolved to compensate for MSCI-induced silencing of *Pdha1*, thereby ensuring male germ cell development.

PDHA1, PDHA2 and PDHB localize to mitochondria in the testis (Figs 1C and 5D,E, Fig. S2A) as peripheral mitochondrial proteins (proteins attached to the inner mitochondrial membrane) (Fig. 5G,H). PDHA1 and PDHA2 are present in the nucleus in addition to mitochondria (Figs 1D and 5F, Fig. S8C), but PDHB is not in the nucleus (Fig. 5D, Fig. S8B). Additionally, interactions between these three proteins were observed in germ cells (Fig. 5C, Figs S6D and S7E). These observations suggest that PDHB binds to both PDHA1 and PDHA2 in mitochondria. Deletion of PDHA2 causes decreased PDHB but does not cause a decrease in the amount of many other mitochondrial proteins (Fig. S8A). Moreover, *Pdha1* expression restored PDHB protein amounts and mitochondrial localization (Fig. 6E,F). Therefore, we propose that a transient PDHA1/PDHA2-PDHB heterotetramer complex forms the E1 component of the PDC in germ cells, similar to the structure in somatic cells (Zhang et al., 2021). Given that the PDC catalyzes the conversion of pyruvate to acetyl-CoA (Gokcan et al., 2022), an essential step for mitochondrial ATP production, the impaired PDC assembly in *Pdha2* KO germ cells likely underlies the observed ATP deficiency (Fig. 5I). This suggests that the PDHA2-PDHB complex serves the same role in testicular mitochondria as the PDHA1-PDHB complex in somatic cells, and is essential for ATP production.

Mechanistically, this energy shortage disrupts ATP-dependent HR during meiotic DSB repair. Foci of RAD51 and DMC1, key ATPase HR proteins, are markedly reduced on chromosomes in *Pdha2* KO spermatocytes (Fig. 4D-F, Fig. S5A). In contrast, the non-ATPase protein RPA2, which functions upstream of the ATPase HR proteins, remains unaffected (Fig. S5A-C). Because both RAD51 and DMC1 perform DNA strand exchange dependent on ATP during the DSB repair process (Ito et al., 2023), it is reasonable that insufficient ATP levels in germ cells (Fig. 5I) cause dysfunctions in RAD51 and DMC1 but no change in their protein amounts (Fig. 4D-F, Fig. S5A-D). This leads to MSH4 instability (Fig. 4D,G, Fig. S5A) and defective crossover formation, as shown by decreased MLH1 foci (Fig. 4B,C) (Cannavo et al., 2020). Energy metabolism has been linked to meiosis (Brower et al., 2007; Varuzhanyan et al., 2019; Schumacher et al., 2024), and our findings suggest that ATP insufficiency impairs DSB repair, leading to meiotic arrest in *Pdha2* KO males. Finally, defective germ cells are removed by apoptosis (Fig. 2F,G). Although whether ATP-deficiency-induced pachytene arrest represents a general mechanism remains unclear, our study reveals a previously unappreciated link between germ cell-specific PDC composition and meiotic DNA repair, highlighting the metabolic dependency of spermatogenesis.

ATP insufficiency is likely the main cause of meiotic arrest in *Pdha2* KO mice. However, the presence of PDHA2 and PDHA1 in the nucleus suggests they have other potential roles. Our results show that both PDHA2 and PDHA1 localize as clustered granules

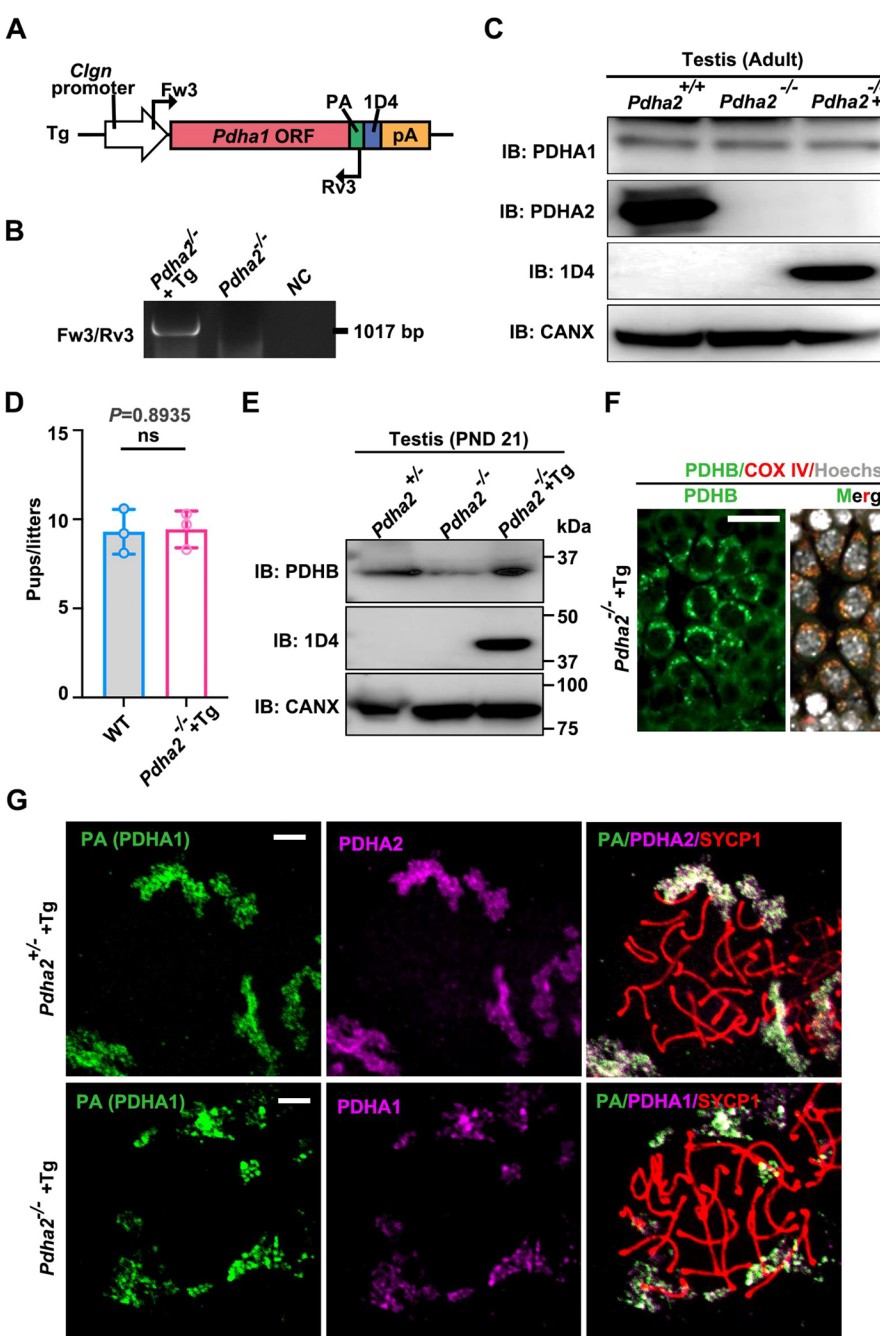

**Fig. 6. Expressed PDHA1 in germ cells after MSCI compensates for PDHA2 function.** (A) Construction of the *Pdha1* transgene. The *Clgn* promoter was used to overexpress *Pdha1* with a PA and 1D4 tag. Black arrowheads indicate primers used for genotyping. (B) Genotyping of the obtained *Pdha1* Tg mice was performed with Fw#3-Rv#3 primers for the Tg allele. (C) WB analysis of PDHA1 and PDHA2 in adult mouse testis. The antibody against 1D4 was used to detect transgene-derived PDHA1. CANX was used as a loading control. IB, immunoblot. (D) Mating test of *Pdha2⁻/⁻*+Tg male mice with three WT females. *n=3* males were used in each group. Data shown as mean±s.d. Statistical analysis by two-tailed, unpaired *t*-test; ns: no significance. (E) WB analysis of PDHB expression in *Pdha2⁺/⁻*, *Pdha2⁻/⁻* and *Pdha2⁻/⁻*+Tg mouse testes at PND 21. Anti-1D4 antibody was used to detect transgene-derived PDHA1. CANX was used as a loading control. (F) PDHB (green), COX IV (red) IF analysis of PND 21 testis in *Pdha2⁻/⁻*+Tg testis. Hoechst (white) was used for visualizing nuclei. Scale bar: 5 µm. (G) Representative nuclear spreads of spermatocytes in the adult testis. Pachytene spermatocytes were stained for SYCP3 (red). Anti-PA (green), anti-PDHA1 (magenta) and anti-PDHA2 (magenta) antibodies were used. Scale bars: 5 µm. Images are representative of three samples.

surrounding the chromosomes (Figs 1D and 6G, Fig. S8C) in pachytene spermatocytes. Their localization pattern is distinct from the typical chromosome-axis localization of meiotic proteins such as HR-related proteins (Fig. 4D, Fig. S5A,B), which indicates an unconventional role in the nucleus. Our localization data is consistent with the reported nuclear expression of PDHA2 and its potential RNA-binding role in meiosis (Li et al., 2024). Another study further implicated PDHA2 in histone modification, likely related to the reduced levels of nuclear acetyl-CoA observed in *Pdha2* KO spermatocytes (Wang et al., 2025). Despite minor differences in staging criteria, both studies consistently reported defects at the pachytene-diplotene transition in *Pdha2* KO mice (Wang et al., 2025). Our IP-MS analysis identified candidate interacting proteins, including PDHX, TARDBP and HDAC6 (Fig. 6B), providing further support for a possible epigenetic role for

PDHA2 during meiosis. Notably, a prior study clarified that pyruvate and the PDC transiently move from mitochondria to the nucleus during zygotic genome activation in mice (Nagaraj et al., 2017). Given the high amino acid sequence similarity (95%) between PDHA1 and PDHA2 (Fig. S7A), it is plausible that PDHA2 may also translocate from mitochondria to the nucleus. Nuclear PDH (PDHA1) has been proposed to be activated without phosphorylation and convert pyruvate to acetyl-CoA to adjust histone modification (Nagaraj et al., 2017). However, its exact role has not been experimentally confirmed. There is currently no evidence about how PDHA2 and PDHA1 directly participate in epigenetic regulation. In this study, we demonstrate that PDHB does not localize to the nucleus, suggesting that PDHA2 and PDHA1 function independently of PDHB within the nucleus. Furthermore, our IF results showed that PDHA2 was absent from the XY body

despite PDHA1 localizing to the XY body (Figs 1D and 5F, Fig. S8C), implying potential distinct nuclear roles for PDHA2 and PDHA1. Since *Pdha1* expression after MSCI rescued PDHB expression (Fig. 6E,F) and fertility in *Pdha2* KO mice, it can be inferred that the primary role of PDHA2 in meiosis might be related to mitochondria. However, we should not overlook PDHA2 localization to the nucleus because this may indicate other functions, such as histone modification and an RNA-binding role (Li et al., 2024; Wang et al., 2025).

In summary, we identify that PDHA2, a mitochondria-associated enzyme, plays a crucial role during meiosis, as reported before. It is known that various testis-specific mitochondrial proteins are associated with energy metabolism, as well as being essential for spermatogenesis (Brower et al., 2007; Varuzhanyan et al., 2019; Schumacher et al., 2024). Many of their disruptions are thought to result in ATP production failure. But the phenotypic consequences of their disruption vary considerably, making it challenging to predict the impact of individual mitochondrial enzyme deficiencies. Our findings indicate that PDHA2 and PDHA1 are temporally co-expressed and can compensate for each other's roles in mitochondria and potentially in the nucleus. This supports the broader hypothesis that autosomal retrogenes derived from X-linked progenitor genes evolved to bypass MSCI, acquiring essential roles in male germline development.

## MATERIALS AND METHODS
### Experimental model and subject details
#### Animals
Animal experiments were conducted using protocols approved by the Institutional Animal Care and Use Committee (IACUC) at the Research Institute for Microbial Diseases of the University of Osaka. Mice were maintained in an environment with stable temperatures, 12 h light/dark cycles, and adequate food and water. B6D2F1 (C57BL/6×DBA2), ICR and C57BL6/J mouse strains were utilized for embryo donation, fostering and RNA extraction, respectively. These strains were obtained from commercial vendors CLEA Japan, Inc. (Tokyo) and Japan SLC (Shizuoka).

#### Generation of *Pdha2* KO mice
The *Pdha2* KO mouse line was generated using CRISPR-Cas9-mediated genome editing by electroporation, as previously described (Abbasi et al., 2018). In brief, two crRNAs, 5′-ACTCATAGACAATTCGATCA-3′ and 5′-ATGCCTGAATCCTACTTAGG-3′ (Fig. 2A), were designed to delete the open reading frame (ORF) of *Pdha2*. To form the Cas9 ribonucleoprotein complex, synthesized crRNAs and tracrRNA (both from Merck) were incubated with Cas9 protein (Thermo Fisher Scientific). The ribonucleoprotein complex was electroporated into 61 fertilized eggs of B6D2F1 using a NEPA21 electroporator (Nepagene). Following electroporation, 56 viable eggs were transplanted into the oviducts of pseudopregnant female mice (ICR). From these transplants, nine mice (F0) were born, of which three pups had the desired mutations in the *Pdha2* gene. Mutant mice were bred with B6D2F1 to generate the next generation. Obtained heterozygous mice were mated to each other to obtain homozygous *Pdha2* KO mice.

#### Generation of *Pdha1* transgenic mice with *Pdha2* KO backgrounds
To generate a transgene construct, PA and 1D4-tagged mouse *Pdha1* cDNA (ENSMUSG00000031299) with a rabbit poly (A) signal under the mouse *Clgn* promoter (Addgene plasmid #173686) (Ikawa et al., 2001) was prepared. The linearized DNA construct was injected into the pronucleus of 184 fertilized eggs with *Pdha2* mutations. After cultivation, 157 two-cell-stage embryos were transplanted into the oviduct ampulla of pseudopregnant ICR mice. Twenty-seven pups were obtained, of which nine pups possessed transgenes. Mice with *Pdha1* transgenes were mated with *Pdha2* KO mice to generate Tg mice on the *Pdha2* KO background. To keep *Pdha1* Tg mice, Tg male mice with *Pdha2* homozygous KO were mated

with female *Pdha2* homozygous KO mice. Genotyping primers are listed in Table S1.

#### Cell lines
COS-7 cells were gifted from Riken BioResource Center (Riken BRC). HEK293T cells were provided by the Verma Laboratory (Salk Institute for Biological Studies, La Jolla, CA, USA). The cell lines were cultured in Dulbecco's Modified Eagle Medium (DMEM, High Glucose, Pyruvate; Thermo Fisher Scientific) supplemented with 10% fetal bovine serum (Biowest). All cell lines were incubated at 37°C in a humidified atmosphere containing 5% $CO_2$. Although the cell lines were not authenticated by short tandem repeat profiling and not regularly tested with *Mycoplasma* detection kits, they were stably maintained by continuous subculturing, exhibited consistent morphology and proliferation, and showed no visible signs of contamination during routine culture.

#### Bacterial strains
This study used the *Escherichia coli* DH5α strain (Toyobo) for molecular cloning procedures. *E. coli* cells were cultured in either lysogeny broth or 2×Yeast Extract Tryptone medium containing 100 mg/l ampicillin. Transformations and cloning were performed by standard heat-shock protocols (Hanahan, 1983).

#### Sequence comparison analysis
Amino acid sequences of PDHA2 were obtained from the NCBI Entrez Protein database. Clustal W2.1 was used for multiple sequence alignment (Larkin et al., 2007). The accession numbers of the PDHA2 proteins used in this study were: PDHA2 mouse (NP_032837.1); PDHA2 human (NP_005381.1); PDHA1 mouse (NP_032836.1).

#### RT-PCR
Total RNA was extracted from multiple adult tissues (*n*=3 biological replicates per tissue) of C57BL6/N mice, and testes from mice aged PND 0-35 (*n*=3 per time point) using TRIzol reagent (Thermo Fisher Scientific). SuperScript IV Reverse Transcriptase (Thermo Fisher Scientific) was used to prepare cDNAs in accordance with the manufacturer's instructions. PCR was performed with KOD FX Neo polymerase (Toyobo). Table S1 provides a summary of the amplification conditions and primers (GeneDesign) for every gene. No samples were excluded.

#### Genotype analysis
PCR was performed with KOD FX Neo. The primers for each gene are listed in Table S1. PCR products were purified using the Wizard SV Gel and PCR Clean-Up System (Promega) according to the manufacturer's instructions. Purified amplicons were then subjected to Sanger sequencing with an ABI 3130xl Genetic Analyzer (Thermo Fisher Scientific) with appropriate sequencing primers. All oligonucleotides used in this study, including PCR and sequencing primers, were synthesized by GeneDesign.

#### Generation of epitope-tagged *Pdha1* and *Pdha2* expression vectors
Full-length coding sequences (ORFs) of mouse *Pdha1* and *Pdha2* cDNA were cloned from C57BL/6J mouse testes by PCR separately. The primers were designed to amplify the entire ORF without untranslated regions, and included a Kozak consensus sequence (GCCACC) upstream of the start codon to enhance translation efficiency. The PCR products were inserted in-frame into expression vectors under the control of either the CAG or *Clgn* promoter using restriction enzyme digestion and ligation. A PA tag and a 1D4 epitope tag with a rabbit globin poly (A) signal were fused to the C terminus of the expressed proteins to facilitate immunodetection. The CAG promoter is a strong, widely used synthetic promoter in mammalian expression vectors, known for driving high gene expression levels. *Clgn* promoter is expressed in male germ cells from the pachytene stage to elongated spermatids (Ikawa et al., 2001). PA tag is derived from the influenza hemagglutinin protein, and the 1D4 tag is an epitope tag corresponding to the last nine amino acids of bovine rhodopsin. We used the anti-1D4 antibody for immunodetection of transfected PDHA1 or PDHA2 in this study. The primers used in this study are listed in Table S1.

## Fertility analysis

The fertility of male mice was evaluated through natural mating tests. Each of the three males were individually housed with three B6D2F1 females for 2 months. The sample size ($n$=3) was chosen based on previous similar studies. Mating plugs were checked daily to record mating frequency and number of pups delivered. No animals were excluded from the analysis. Males and females were randomly assigned to cages. The investigator was unaware of genotype during pup counting.

## Morphological and histological analysis of testis and epididymis

Testes and epididymides of adult *Pdha2* KO males aged 12 weeks were obtained by dissection after euthanasia by cervical dislocation or $CO_2$ inhalation. Testicular weights and body weights from three males were recorded. The testes and epididymides were fixed with Bouin's solution (Polysciences). Fixed samples were dehydrated, embedded in paraffin, and sectioned at 5 μm thickness. Sections were then treated with 1% periodic acid for 10 min, Schiff's reagent (Wako) for 20 min, and stained with Mayer's Hematoxylin solution. The slides were washed three times, and coverslipped with Immu-Mount (Thermo Fisher Scientific). The sections were observed using a BZ-X710 microscope (Keyence).

## Apoptosis detection in testicular sections

Apoptotic cells in seminiferous tubules were detected using the TUNEL method following the manufacturer's protocol with the In Situ Apoptosis Detection Kit (Takara Bio Inc.). In brief, testes were fixed with Bouin's fixative, embedded in paraffin, and sectioned at 5 μm. After paraffin removal, antigen retrieval was performed in citric acid buffer (pH 6.0) at 95°C for 20 min. Endogenous peroxidase activity was quenched by incubating sections in 3% $H_2O_2$ at room temperature (RT) for 5 min. Sections were then incubated with TdT enzyme and FITC-conjugated dUTP at 37°C for 1 h to label DNA-strand breaks. For visualization, sections were incubated with HRP-conjugated anti-FITC antibody at 37°C for 1 h, followed by 3 min incubation with ImmPACT DAB substrate (Vector Laboratories). After washing, sections were counterstained with Mayer's Hematoxylin for 1 min, dehydrated through an ethanol gradient, and mounted with Permount (Ferma). Stained sections were examined using a BX53 microscope (Olympus).

## Generation of antibodies

Rabbit polyclonal antibodies were produced by immunization with mouse PDHA1 (NP_032836.1) polypeptide (C plus DPPFEVRGANQW LKFKSVS), mouse PDHA2 (NP_032837.1) polypeptide (C plus EVRGAHKWLKYKSHS) or PDHB (NP_077183.1) polypeptide (C plus EKVFLLGEEVAQYDGAYKV). These antibodies were purified from serum using their polypeptide and SulfoLink coupling resin (Thermo Fisher Scientific).

## Immunoblotting

To extract proteins, testes and HEK293T cells were homogenized in T-PER lysis buffer (Thermo Fisher Scientific) supplemented with a 1:100 dilution of a protease inhibitor cocktail. Liver tissues were collected following perfusion to remove blood under anesthesia, and subsequently lysed using the same procedure as for the testes. Proteins were reduced with 2-mercaptoethanol at 95°C incubation, separated by sodium dodecyl sulfate polyacrylamide gel electrophoresis (SDS-PAGE), and then transferred to polyvinylidene fluoride (PVDF) membranes using the Trans-Blot Turbo system (Bio-Rad). Membranes were blocked with 10% skimmed milk (Becton Dickinson). After incubating with primary antibodies overnight at 4°C, washing was carried out with TBST (PBS containing 0.05% Triton X-100) three times. Next, membranes were incubated with HRP-conjugated secondary antibodies for 2 h at RT. Chemiluminescence was detected with Chemi-Lumi One Super (Nacalai Tesque) using the Image Quant LAS 4000 mini (GE Healthcare). See Table S2 for a detailed list of primary and secondary antibodies used in this study.

## Subcellular fractionation and mitochondrial assays

Subcellular fractionation and mitochondrial assays were conducted as previously described (Mise et al., 2022). WT testes were suspended in a homogenization buffer (250 mM sucrose, 20 mM HEPES-NaOH, pH 7.5) and homogenized with a Dounce homogenizer. The homogenate was centrifuged at 1000 $g$ for 10 min at 4°C to remove debris, and the resulting supernatant was further centrifuged at 10,000 $g$ for 10 min at 4°C. The resulting mitochondrial pellet was suspended in 20 mM HEPES-NaOH (pH 7.5).

For sodium carbonate extraction, suspended mitochondria were subjected to ultrasonic treatment, incubated with or without 0.1 M $Na_2CO_3$ (pH 12.0) for 45 min on ice, and centrifuged at 100,000 $g$ for 30 min at 4°C. The resulting supernatant and pellets were subjected to immunoblot analysis.

For the proteinase K digestion assay, the suspended mitochondria were incubated for 20 min at 37°C with or without proteinase K (final concentration 0.8 mg/ml), and 1% Triton X-100. The reaction was terminated by the addition of phenylmethylsulfonyl fluoride (final concentration 2.0 mM), followed by the addition of SDS sample buffer before immunoblot analysis.

## ATP luciferase assay

Before the ATP luciferase assay, a single-cell suspension was performed to remove all cells except germ cells according to a previous study with slight modifications (Gaysinskaya et al., 2014). Male mice (PND 21) were euthanized by cervical dislocation or $CO_2$ inhalation and the testes were dissected. After tunica albuginea removal, the testes were incubated in DPBS containing 0.5 mg/ml collagenase and 5.0 μg/ml DNase I at 34°C with high-speed agitation for 8-10 min. The supernatant was filtered through a 70 μm nylon cell strainer. The remaining pellets were further digested in DPBS containing 0.25% trypsin and 2.5 μg/ml DNase I at RT with gentle agitation and pipetting for 30 min. Fetal bovine serum (final concentration 10%) was added to terminate digestion. The dissociated cells and the supernatant were gathered in a collecting tube through a 40 μm strainer cap. The strained solution was used as single-cell suspension of germ cells.

After single-cell suspension, germ cell concentration was calculated with a one-cell counter (Biomedical Science). The ATP level of germ cells ($3.0 \times 10^5$ cells) was measured using an ATP Detection Assay Kit (700,410, Cayman Chemical) following the manufacturer's instructions. Luminescence was measured with a multi microplate reader GloMax Multi+ Detection System (Promega).

## Immunoprecipitation

Protein lysates from seminiferous tubules were extracted using NP40 lysis buffer. Protein lysates were mixed with Dynabeads Protein G (Thermo Fisher Scientific)-conjugated antibodies. The immune complexes were incubated for 1 h at 4°C, and washed three times with wash buffer. Co-immunoprecipitated products were then eluted with SDS sample buffer, and incubated for 10 min at 70°C. The antibodies used in this study are listed in Table S2.

## MS and data analysis

Immunoprecipitated proteins were reduced, alkylated and digested. The resulting peptides were purified using C18 tips (GL-Science). Peptides were analyzed by nanocapillary reversed-phase LC-MS/MS using a C18 column (25 cm×75 μm, 1.6 μm; IonOpticks) on a nanoLC system (Bruker Daltoniks) coupled to a timsTOF Pro mass spectrometer (Bruker Daltoniks) equipped with a CaptiveSpray ion source.

The resulting data was processed using DataAnalysis version 5.1 (Bruker Daltoniks). Protein identification was performed using MASCOT version 2.7.0 (Matrix Science) against the SwissProt database. Quantitative analysis and fold change calculations were carried out with Scaffold5 (Proteome Software).

## Ontology analysis for proteins identified from MS analysis

Candidate interacting proteins were identified by MS. To explore their potential biological functions and associated pathways, GO and Kyoto Encyclopedia of Genes and Genomes (KEGG) enrichment analyses were conducted using Metascape (https://metascape.org) with default parameters (Zhou et al., 2019). The enrichment results were subsequently processed using custom scripts to extract the top 15 pathways ranked by the lowest logP values. These pathways were visualized in the form of a bubble plot to illustrate relative significance and gene count.

## Expression pattern of mRNA and proteomic level in spermatogenesis

For the mRNA expression pattern, we used previously published data sets, queryable single-cell RNA-sequencing datasets of human and mouse spermatogenic cells (Hermann et al., 2018). For the protein expression pattern in germ cells, we analyzed data from a published paper (Fang et al., 2021). ATP5A1 (or ATP5F1A) was used as a standard for relative expression as it showed stable expression during spermatogenesis.

## IF analysis using testis cryosections and cultured cells

The testes were fixed overnight at 4°C in 4% paraformaldehyde (PFA), then transferred through a graded series of sucrose in PBS, embedded in OCT compound (Sakura Finetek), and 8.0 μm sections were cut with a cryostat (CryoStar NX70, Thermo Fisher Scientific). Antigen retrieval was performed in citric acid buffer (pH 6.0) at 95°C for 20 min. Samples were then permeabilized with 0.1% Triton X-100 in PBS for 20 min, and blocked with 3% bovine serum albumin (Merck) in PBS for 30 min. The samples were incubated with primary antibody overnight at 4°C in a blocking buffer, and incubated in appropriate Alexa Fluor-conjugated secondary antibodies with Alexa Fluor-conjugated lectin PNA (Thermo Fisher Scientific) for 2 h at RT. The sections were stained with Hoechst 33342 for visualizing nuclei, and then coverslipped with Immu-Mount.

COS-7 cells ($1.5 \times 10^5$ cells) were seeded on coverslips in a 6-well plate. After 6-8 h, expression vectors were transfected into the cells using PEI MAX (Polysciences). The cells were fixed with 4% PFA 40 h after transfection, and permeabilized with 0.5% Triton X-100 in PBS. The protocol thereafter was the same as for the testis cryosections described above.

Microscopic and fluorescence images were taken with a Nikon Eclipse Ti microscope connected to a C2 confocal module (Nikon). Fluorescence images were false-colored and cropped using ImageJ software (version 2.0.0, NIH). Table S2 provides a detailed list of primary and secondary antibodies used in this study.

## IF analysis using chromosome spreads from mouse spermatocytes

Chromosome spreads from mouse spermatocytes were performed as previously described (Oura et al., 2021). Briefly, testes were incubated with hypotonic buffer (30 mM Tris, pH 8.2, 50 mM sucrose, 17 mM trisodium citrate dihydrate, 5 mM EDTA, pH 8.0, 2.5 mM dithiothreitol, 0.5 mM phenymethylsulfonyl fluoride) for 40 min at RT after tunica albuginea removal. The samples were then transferred to 100 mM sucrose, and germ cells were released by gently squeezing the seminiferous tubules. Germ cells were fixed in 1% PFA with 0.15% Triton X-100. The cell suspension was evenly spread on the slides, followed by incubation in a humid chamber for 2 h at RT. Then washed in 0.04% PhotoFlo solution to remove salts (Kodak Alaris).

For IF, chromosome spreads on slides were blocked with 3% non-fat milk in PBS for 10 min, then incubated with primary antibodies in 3% non-fat milk at 37°C overnight. Slides were washed three times in PBS with 0.1% Tween 20, and incubated with secondary antibodies for 1 h at 37°C. The slides were washed three times, and coverslipped with Immu-Mount. Images were captured with an FV3000 confocal microscope (Olympus). Table S2 provides a detailed list of primary and secondary antibodies used in this study.

All images were processed using Adobe Photoshop CC (Adobe Systems, Inc.). We identified the XY bodies as chromosomes with a condensed, asymmetrical SYCP3 configuration with partial synapsis, distinct from linear and fully synapsed autosomes. We distinguished different stages of pachytene spermatocytes according to H1t signal, sex chromosome morphology, and autosome length/shape by SYCP3 staining. Details have been previously described (Alavattam et al., 2018). Single cells were manually cropped and analyzed with semi-automated scripts in Fiji (Boekhout et al., 2019) (https://github.com/Boekhout/ImageJScripts). Briefly, for quantification of foci, images were auto-thresholded on SYCP3 staining as a mask using 'Find Maxima' to determine the number of RPA, DMC1, RAD51, MSH4 and MLH1 foci. Manual inspection of the images ensured there were no obvious defects in identifying the SYCP3 axis. This included counting axes from neighboring cells, detecting artifacts, or missing clear foci from the script. All images were manually inspected for analysis. For some staining results with high background, such as RAD51 and MSH4, we rotated the red channel (SYCP3) 90° to remove overlapping signals and reduce errors.

## Image selection and quantification criteria

For quantification of recombination foci in spermatocyte chromosome spread, only cells with well-spread chromosomes, clearly defined SYCP3 axes, and unambiguous meiotic staging were included. In RAD51 immunostaining of *Pdha2* KO spermatocytes, a small subset of cells exhibited disorganized or aggregated RAD51 signals that did not colocalize with properly spread SYCP3 axes. These cells appeared to retain clustered chromatin, possibly due to incomplete nuclear envelope breakdown or defective chromatin dispersion. Although they were H1t positive, indicating entry into the mid-pachytene stage, the chromosomes were abnormally short or fragmented, rendering both staging and quantification of foci unreliable. Such cells were excluded based on pre-established morphological and signal quality criteria (Alavattam et al., 2018). These exclusions represented a small fraction of the total analyzed and did not affect the overall conclusions. All images were manually inspected following semi-automated quantification to ensure consistency. No formal randomization or anonymization of groups was applied, but uniform image acquisition and processing was used to minimize potential bias.

## Statistical analysis and reproducibility

All statistical analyses were conducted using GraphPad Prism software (version 9.5.1, GraphPad Software). Bars in figures indicate mean±s.d., with individual data points plotted for each biological replicate unless otherwise specified. Specific statistical parameters, tests employed and sample sizes are detailed in the respective figure legends. Sample sizes were not predetermined using statistical methods. *In vivo* experiments were typically conducted on at least three animals per genotype unless otherwise noted, and consistent phenotypic outcomes were observed across biological replicates. For biochemical analyses, three independent experiments were performed to ensure reproducibility. Investigators were aware of group allocation during the data analysis. For comparisons between two groups, an unpaired, two-tailed Student's *t*-test was used. Investigators were aware of group allocation; however, objective quantitative analyses, such as IF foci counting, were performed using standardized protocols and semi-automated software tools to minimize observer bias. Statistical significance was typically set at $P<0.05$, with specific $P$-values and significance levels indicated in the figures and figure legends.

### Acknowledgements

We also thank the members of the Central Instrumentation Laboratory, Research Institute for Microbial Diseases, The Universrity of Osaka, especially Keiko Murata for Sanger sequencing, and Akinori Ninomiya and Fuminori Sugihara for mass spectrometry analyses. We thank the members of the Department of Experimental Genome Research, Animal Resource Center for Infectious Diseases, and NPO for Biotechnology Research and Development for their experimental assistance. We sincerely appreciate Ewelina Bolcun-Filas (The Jackson Laboratory) for providing us with H1t antibodies, and gratefully thank Masaru Ito (The University of Osaka) for supporting us with the DMC1 antibody. We also thank Julio Castaneda (College of Southern Idaho) for critical reading. We appreciate Kiyonori Kobayashi, Chihiro Emori, Taichi Noda and Nobuyuki Sakurai for technical assistance.

### Competing interests

The authors declare no competing or financial interests.

### Author contributions

Conceptualization: K.S., M.I.; Data curation: C.P., K.S., H.-Y.C., H.W.; Formal analysis: C.P., K.S.; Writing – original draft: C.P., K.S.; Writing – review & editing: C.P., K.S., H.-Y.C., H.W., M.I.

### Funding

This work was supported by KAKENHI grants from the Ministry of Education, Culture, Sports, Science and Technology/Japan Society for the Promotion of Science (JP23K05831 to K.S.; JP21H05033, JP23K20043 to M.I.); an Adopting Sustainable Partnerships for Innovative Research Ecosystem (ASPIRE) grant from Japan Science and Technology Agency (JP23jf0126001 to M.I.); a Japan Science and Technology Agency grant (JPMJCR21N1 to M.I.); Takeda Science Foundation grants (to K.S. and M.I.); a Senri Life Science Foundation grant (to K.S.); the Eunice Kennedy Shriver National Institute of Child Health and Human Development

(R01HD088412 to M.I.); and OU Master Plan Implementation Project promoted under the University of Osaka. Open Access funding provided by the Eunice Kennedy Shriver National Institute of Child Health and Human Development. Deposited in PMC for immediate release.

**Data and resource availability**
Frozen spermatozoa from *Pdha2* KO mice and *Pdha1* Tg mice on the *Pdha2* KO background were deposited under the names B6D2-*Pdha2*[em1Osb] and B6D2-*Pdha2*[em1Osb] Tg (Clgn-*Pdha1*/PA/1D4)1Osb, respectively. These mouse lines are available through the Riken BioResource Center (Riken BRC) or the Center for Animal Resources and Development, Kumamoto University (CARD). The stock ID number of *Pdha2* KO mouse strain is 11230 (Riken BRC) or 3034 (CARD), and the ID of *Pdha1* Tg mouse on the *Pdha2* KO background is 12230 (Riken BRC) or 3461 (CARD), respectively.

**The people behind the papers**
This article has an associated 'The people behind the papers' interview with some of the authors.

**Peer review history**
The peer review history is available online at https://journals.biologists.com/dev/lookup/doi/10.1242/dev.204683.reviewer-comments.pdf

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
