## [Peer Review File · Development (Cambridge, England)]

Compensation for X-linked *Pdha1* silencing by *Pdha2* is essential for meiotic double-strand break repair in spermatogenesis

Chen Pan, Keisuke Shimada, Hsin-Yi Chang, Haoting Wang and Masahito Ikawa

DOI: 10.1242/dev.204683

Editor: Haruhiko Koseki

Review timeline

Original submission:	28 January 2025
Editorial decision:	13 March 2025
First revision received:	29 May 2025
Editorial decision:	30 June 2025
Second revision received:	2 July 2025
Accepted:	2 July 2025

Original submission

First decision letter

MS ID#: dev.204683

MS TITLE: Compensation for X-linked *Pdha1* silencing by *Pdha2* is essential for meiotic double-strand break repair in spermatogenesis

AUTHORS: Chen Pan, Keisuke Shimada, Hsin-Yi Chang, Haoting Wang and Masahito Ikawa

Dear Dr Shimada,

I have now received all the referees' reports on the above manuscript, and have reached a decision. The referees' comments are appended below, or you can access them online: please go to:

As you will see, the referees express considerable interest in your work, but have some significant criticisms and recommend a substantial revision of your manuscript before we can consider publication. If you are able to revise the manuscript along the lines suggested, which may involve further experiments, I will be happy receive a revised version of the manuscript. Your revised paper will be re-reviewed by one or more of the original referees, and acceptance of your manuscript will depend on your addressing satisfactorily the reviewers' major concerns. Please also note that Development will normally permit only one round of major revision. If it would be helpful, you are welcome to contact us to discuss your revision in greater detail. Please send us a point-by-point response indicating your plans for addressing the referees' comments, and we will look over this and provide further guidance.

Please attend to all of the reviewers' comments and ensure that you clearly highlight all changes made in the revised manuscript. Please avoid using 'Tracked changes' in Word files as these are lost in PDF conversion. I should be grateful if you would also provide a point-by-point response detailing how you have dealt with the points raised by the reviewers in the 'Response to Reviewers' box. If you do not agree with any of their criticisms or suggestions please explain clearly why this is so.

Reviewer 1

Advance summary and potential significance to field

The manuscript by Pan et al. investigates the role of PDHA2 in spermatogenesis. They find that PDHA2 is crucial for maintaining efficient DSB repair and proper meiotic progression in male germ cells, likely through mitochondrial ATP production. This study also supports an interesting hypothesis that testis-specific retrogenes compensate for the silencing of X-linked progenitor genes during MSCI. Overall, this manuscript is very well written with insightful discussion. The story about the function of PDHA2 in mitochondria for DSB repair is generally convincing, and the data are of high quality. This paper should be of interest to the readers in the field of germ cell biology and meiosis. Meanwhile, I would like to suggest including additional descriptions of the data, particularly for the nuclear localization and function of PDHA2, which are necessary for supporting their important conclusions.

Comments for the author

1. This study used *Pdha2*^{+/-} as a control for *Pdha2*^{-/-}. A description of whether *Pdha2*^{+/-} males are completely normal is needed.
2. Supplementary Figure 1C: A TOM20-only panel should also be included to show mitochondrial signals independently.
3. Page 8, lines 164-176: The text appears to be jumbled, likely due to a file error.
4. Page 6, line 129 states, "when we housed *Pdha2*^{+/-} and *Pdha2*^{-/-} males with three WT females for 2 months separately, no pups were sired from three independent males." Pups were obtained from control males (Fig. 2D) - please correct this mistake. Figure 2D is labeled ^{+/+} and ^{-/-}. Please confirm which genotype was used as the control.
5. Although nuclear signals of PDHA2 are very weak in section IF images, the authors suggest nuclear localization of PDHA2 based merely on spread IF results showing strong signals. Careful evaluation is needed to draw such a conclusion because cytoplasmic proteins can come into contact with nuclear components upon disruption of cellular membranes in spread experiments. Can the authors exclude the possibility that the signals are derived from cytoplasmic PDHA2?
6. Although PDHA1 is localized in XY bodies in the nucleus, the spread IF results suggest that PDHA2 is not associated with XY bodies. This indicates that their roles in the nucleus may be distinct. Since PDHA1 can compensate for the loss of PDHA2 during spermatogenesis, it can be inferred that the primary role of PDHA2 in meiosis is related to mitochondria. This point should be included in the discussion.
7. Page 10, line 218: Please indicate what tissues/cells from which stage of mice were used for the IP experiment.
8. How the authors identified XY bodies in spread IF experiments should be explained in the Methods and/or legends.
9. Fig. 5B: Symbols in the *Pdha2* KO column should be explained in the legend.
10. It may be useful to include gene ontology analysis for the proteins identified from the MS-spec analysis.
11. Because the major highlights of this paper are the localization and function of PDHA2 in mitochondria and the nucleus for meiosis, they should include a description of what nuclear proteins were identified from the MS-spec analysis, if any, and provide a discussion. Were RNA-binding proteins included?
12. Throughout the manuscript, exact P-values should be shown in figures rather than asterisks/ranges.

Reviewer 2*Advance summary and potential significance to field*

This work concerns the testis-specific peripheral mitochondrial protein *Pdha2* (pyruvate dehydrogenase E1 alpha 2). It is postulated that *Pdha2* on mouse chromosome 3 is an autosomal retrogene of *Pdha1*, which is X-linked. Deletion of *Pdha2* results in male infertility (azoospermia) due to failure in spermatogenesis at the late pachytene-diplotene transition of meiosis. Interestingly, the phenotype can be rescued by expression of *Pdha1* from an autosome (the X-linked *Pdha1* would be silenced during meiotic sex chromosome inactivation (MSCI)). Results suggest that PDHA2 forms the pyruvate dehydrogenase complex in mitochondria by interacting with PDHA1 and

PDHB, and this is required for normal levels of ATP (for function of ATPase recombinase proteins RAD51 and DMC1) in male germ cells. As RAD51 and DMC1 direct formation of crossovers, their loss of function is associated with defective double-strand break repair during pachytene; the defective germ cells are removed by apoptosis.

A constitutive Pdha2 KO was made, the validity of this was demonstrated, and the phenotype was thoroughly and carefully analysed, creating large amount of data. Although a similar study has very recently been published (Wang et al, Cell Proliferation, Feb 2025) the manuscript under consideration is still an important contribution. One part of the current study that adds quite a lot, in my opinion, is that the authors were able to demonstrate, in vivo, that germ cell-specific PDHA2 can be compensated by PDHA1. PDHA1 is used in most cell types, but can not be produced once germ cells enter meiosis, due to MSCI. This study is a good complementary study to the Want et al, 2005 study mention above - together they substantially expand our understanding of the role of the pyruvate dehydrogenase complex (PDC; links glycolysis to the oxidative TCA cycle) in spermatogenesis. The work demonstrates an excellent analysis of the progression of spermatogenesis, with a broad range of characteristic and relevant antibodies used. The complicated process and the roles of the various proteins visualized are well explained. I imagine others will use this work as a template as they analyse mouse models with arrest at a similar stage.

Comments for the author

Some questions for the authors to address:

1. I was surprised that PDHA2 interacts physically with PDHA1. It seems more likely that PDHA2 is just required in the special circumstances of MSCI, to compensate for PDHA1 (and complex with PDHB). I think this result might reflect that PDHA proteins normally interact with PDHB to form a heterotetramer (line 79) - so PDHA1 and PDHA2 can interact, though this would not normally be seen in vivo, because they would not normally be co-expressed? Maybe the authors could clarify this point. If I am curious then I presume other readers would be too.
2. Why would the amount of PDHB decrease when there is no PDHA2? Could it be that it is not stable in the absence of a complexing partner?
3. I wonder whether the title should reflect more about the evolutionary gene pair and what has been revealed regarding functional compensation during MSCI
4. Line 308 - 'remarkable reductions in ATPase proteins (RAD51 and DMC1) expression are observed'. Is this a reduction in expression/amount of protein though, or a failure to localize? We would probably expect that the protein is made normally, but does not localize to foci, so is not clearly observable?

Additional minor points for the authors to address:

1. I had no idea what a 'peripheral' mitochondrial protein is, and it is likely others will also not know.
2. Line 68 - sentence beginning here is not complete.
3. Line 107 - it would be useful to clarify what PA and 1D4 are (as not all readers would be making constructs). This could be in the appropriate figure legends or in the methods.
4. Line 158 - 'MSCI is initiated by BRCA1 localization in early pachytene spermatocytes' - please include a reference here.
5. Line 164 - from here numerous errors in the text.
6. Line 278 - 'testis-specific Clgn promoter' - please include a reference here.

7. Figure 1 legend. Please include info that Fig1B shows PND testis samples (you just show numbers). Mention in the legend that COX IV is highlighting mitochondria. What is the dotted white line?
8. Figure 3 legend. Please define DDR and MSCI.
9. Figure 6 legend. 'compensates' in the title
10. Supp figure 1 legend. Maybe indicate TOM20 is a mitochondrial marker
11. Figure 2. Maybe include the word 'deletion' alongside -2,374 bp
12. Figure 4 . Typo of 'chckpoint'
13. Figure 5. What is the legend (different colours and shapes for PDHB, PDHA2 etc) in figure 5B referring to? Presumably these are left over from previous iteration where Supp Fig6B was included alongside.
14. Figure 6G. Is this labelled correctly (PDHA2 instead of PDHA1 in the upper control panel).

Reviewer 3

Advance summary and potential significance to field

Pan et al present a detailed characterization of the meiotic arrest associated with a loss of PDHA2, a conserved retrogene copied from the X-linked gene PDHA1. PDHA1 is an essential component of the mitochondrial pyruvate dehydrogenase complex (PDC) that is inactivated by MSCI in spermatocytes. The authors show that in mice lacking PDHA2 there is a block in late pachytene prior to diplotene resulting from a failure to correctly repair double strand breaks (DSB). They further show that PDHA2 interacts with PDHA1 and PDHB, components of the PDC, and that ectopic PDHA1 expression in spermatocytes complements for the loss of PDHA2, evidence that PDHA2 functions similarly to PDHA1. In spermatocytes lacking PDHA2, they find reduced levels of ATP and a reduction of DNA repair ATPase proteins RAD51 and DMC1, as well as MLH1 foci (crossovers) on the chromosomes in mid-late pachytene spermatocytes. Finally, the authors present evidence that PDHA1 and PDHA2 localise to the nucleus in pachytene spermatocytes. Although determining the functional significance of this observation will not be easy, it could inspire new avenues of exploration.

The study has been thoroughly performed and the images presented are convincing. An interesting story has been told, in particular around the functional complementarity of PDHA1 and its retrogene PDHA2 - based on both a genetic and proteomics approach. Nevertheless, the article is harder to understand than it need be for someone who does not regularly attend the authors' lab meetings. I have indicated areas that were problematical for me, but the authors should make an effort to revise the text to render it as readable as possible. Despite the publication two weeks ago of an independent article describing the effects of PDHA2-loss on meiotic progression, the present article would make a significant contribution to the field since it adds clean informative proteomics data and a demonstration that PDHA1 complements the loss of PDHA2. A comparison of perceived phenotypic differences in Pdha2-KO mice may also be illuminating.

Comments for the author

The finding that PDHA1 and 2 localise to the nucleus is unexpected for mitochondrial proteins (could the authors add a clear statement as to whether this has already been described or not) especially since their partner PDHB does not. I am not challenging this result, which looks convincing for PDHA1, but the authors should make more effort to show this to be a secure finding for PDHA2. Ideally the specificity and affinity of the antibody raised by the authors for this study against PDHA2 must be validated for the meiotic spreads. For the anti-PDHA2, this should be done on PDHA2+/-, PDHA2-/- + Clgn-PDHA1-tg, and PDHA2-/- mice. This would also show whether there is any cross-reaction with PDHA1. Furthermore, Figure 5 seems to show nicely that PDHA1 signal

colocalizes with gamma-H2AX (XY body), but this is not evident in supp Figure 2A for PDHA2. This must be discussed.

On a related point, there is some confusion in the labelling of Figure 6G. The upper central and upper right panel do not accord in their PDHA labelling - is the magenta PDHA1 or PDHA2?

I was initially confused by the finding that PDHA1 is a potential partner of PDHA2. This suggested either that normally PDHA1 and PDHA2 can be part of the same PDC in spermatocytes, or that each of their anti-PDHA antibodies detects both PDHA1 and PDHA2. I discovered, by literature search, that PDHA1 appears to form homodimers, and that its heterotetramer with PDHB, to give the pyruvate dehydrogenase E1, exists in multiple copies within each PDC. This information is extremely important for the interpretation of this study and it will greatly improve the understanding of this article and the perceived quality of the results for scientists who are not familiar with the PDC. The structure of the PDC must be more fully presented in the Introduction, or at least discussed in the context of the results.

Antibody validation. The authors have generated three antibodies against three of the key proteins in this study, PDHA1, PDHA2 and PDHB. Information must be provided in Material and Methods concerning the strategy for immunogen choice (especially for PDHA1 and 2), immunogens, affinity validation and demonstration of a lack of cross-reaction with PDHA1 or PDHA2. The validation of the anti-PDHA2 specificity in denaturing conditions has been convincingly done by the Western analyses of the *Pdha2*-KO mice. At face value, the IP results suggest that there is equally little cross-reaction between PDHA1 and 2 under non-denaturing conditions. The IF in Supp Fig 2B shows that the anti-PDHA2 certainly has low affinity for PDHA1. A full length Western for each of the three new antibodies should be shown in a supp Figure.

Please state clearly and as precisely as possible the mouse strain or hybrid used in the two different sets of transgenic experiments, taking into account the breeding strategy used to generate the different PDHA genotypes studied. The only information provided is "B6D2F1 (C57BL/6 \bar{A} - DBA2), ICR, and C57BL6/J mouse strains were utilized for embryo donation, fostering, and RNA extraction, respectively." at line 563.

Please present the criteria used to classify meiotic spreads as mid- or late-pachytene.

Please discuss the independent article that has just been published (Fang et al 2025) describing the phenotype associated with loss of PDHA2 function in the mouse. In particular the difference in interpretation of the point of arrest (late pachytene vs diplotene). If possible, state whether or not abnormal diplotene cells are present in the model presented here.

Please state in the results that spermatocytes were enriched by flow sorting prior to measuring ATP levels. This information should be added to the Methods section. An estimate of purity should be added.

The mitochondrial localization experiment presented in Figure 5G and H is poorly explained in the results - lines 252-258.

- "but transmembrane proteins, [such as VDAC1], appear in the pellet."
- "This treatment changes the localization of" is not clear. You could state the observed change.
- "immunoblot signals comparable to those of UQCRC1....." A little introduction to UQCRC1, a protein attached to the inner membrane, is required here.

Minor comments

In light of Fang et al 2025, I would recommend re-working the title to add the original finding that PDHA1 can complement for the loss of PDHA2.

Line 25: I do not think you can say that PDHA2 serves the same roles as PDHA1 in somatic cells. To do this would require you to complement the loss of PDHA1 with PDHA2. You can however say that PDHA1 is able to perform the same roles as PDHA2 in the mitochondria of spermatocytes. You could move "and is essential for [normal] ATP production in pachytene spermatocytes" to the second highlight.

Line 120: >>> Therefore we speculate that PDHA2 may play a role in the nucleus as well as the mitochondria during male meiosis.

Line190: [Compared to controls,] there were no differences in HORMAD1 localisation on X and Y".

Line 317: "A previous study suggested" >>> Previous studies suggested

Line 347: "We hope the analysis"". The sense of this sentence is not obvious.

Line 343: "This finding supports the hypothesis.....". No reference is provided. It is more logical to propose that the creation of the PDHA2 retrogene allowed the silencing of PDHA1 by MSCI.

Please add the age of the mice used in the Figures, where space permits. Otherwise make sure this is clearly mentioned in the Figure legend.

Reviewer 4

Advance summary and potential significance to field

This study by Pan et al. systematically investigates the role of testis-specific PDHA2 in male meiosis using mice models. The authors demonstrate that PDHA2 interacts with PDHB in testicular mitochondria to regulate mitochondrial functions. PDHA2 deficiency leads to ATP depletion in germ cells, impairing ATPase-mediated recombination protein functions and causing insufficient crossover formation during pachytene stage. Consequently, spermatocytes arrest at the pachytene stage and undergo apoptosis, ultimately resulting in azoospermia. Transgenic Pdha1 expression rescued fertility in Pdha2 KO (knock-out) mice, supporting the evolutionary hypothesis of PDHA2's origin from PDHA1 while explaining the pachytene-specific defects. The study is logically structured with solid evidence, advancing our understanding of mitochondrial proteins in spermatogenesis and offering therapeutic insights for PDHA2-related human infertility.

Comments for the author

Major Comments

1. ATP restoration validation: While the ATP depletion mechanism is clearly demonstrated, did the authors attempt to supplement ATP in Pdha2-KO testes to recover male fertility?
2. Metabolic compensation: PDHA2 is essential for mitochondrial function. Were alternative energy substrates (e.g., fatty acids or glutamine) examined for potential compensatory effects on spermatogenesis of this KO?
3. Mitochondrial consequences: Figure 5D/E and Supplementary Figure 2B seems show mitochondrial morphological changes. Were mitochondrial dynamics or oxidative phosphorylation proteins analyzed by WB?
4. PDHA1/2 specificity:
 - The hypothesis that PDHA2 originated from PDHA1 would be strengthened by including a DNA sequence alignment between the two genes, particularly in regulatory regions.
 - Supplementary Fig. 6B/7B indicate that X-linked PDHA1 is silenced during MSCI. Does the Clgn promoter-driven transgene presumably expresses PDHA1 post-MSCI? How does the Tg bypass MSCI-mediated silencing?
5. Nuclear function clarification: Figure 6 and Supplementary Fig. 6 shows that transgenic Pdha1 restores both mitochondrial and nuclear localization. It seems PDHA1 persists in the nucleus beyond mid-pachytene. Does PDHA2-KO impair nuclear-specific functions? This observation strengthens the rescue mechanism and warrants discussion.

Minor Comments

1. Figure issues:
 - Figure 3B/D: Labels appear inconsistent between panels and legends ("WT" vs "Hetero")
 - Figure 4E: Missing annotation

2. Line 71-73: Rephrase metabolic demand statement to: "This highlights the heightened metabolic requirements of meiotic cells, suggesting energy deficits during this phase critically disrupt spermatogenesis."
3. Line 81-83: Clarify unresolved mechanistic links.
4. Line 130: Replace "three independent males" with "KO mating cages".
5. Line 159, 162, 164, 166, 168, 172, 174, 195: Address irregular spacing and citation formatting (reference insertion error).
6. Line 163: Remove redundant "IF".
7. Line 167-169: Revise to correct multiple format and description errors.
8. Line 173: Remove redundant parentheses. And delete incomplete sentence: "During MSCI, the sex chromosomes are transcriptionally."
9. Line 174-175: Consolidate redundant mentions of H3K9me3.
10. Line 299-301: Rewrite for clarity.
11. Line 578: Delete extraneous "-" after PA.
12. Line 794: Correct formatting error.
13. Line 807: Ensure consistent italicization of "N".

First revision

Author response to reviewers' comments

General comments:

We sincerely thank the reviewers for their valuable comments, which have significantly improved the quality of our manuscript. In response, we have made substantial revisions, particularly in the Discussion section. A major focus of our revision was the validation of the antibodies for PDHA1 and PDHA2, given the high sequence similarity (95%) between the two proteins. In accordance with the journal's formatting guidelines, we have also made numerous detailed adjustments throughout the manuscript; these are not individually highlighted. Additionally, we have incorporated all requested sections, with new content highlighted in blue in main text. As per the editor's instructions, we have refrained from using the "Track Changes" function and instead marked all textual revisions in red.

Response to the reviewers:

Reviewer 1: SUMMARY OF THE ADVANCE MADE IN THIS PAPER AND ITS POTENTIAL SIGNIFICANCE TO THE FIELD

The manuscript by Pan et al. investigates the role of PDHA2 in spermatogenesis. They find that PDHA2 is crucial for maintaining efficient DSB repair and proper meiotic progression in male germ cells, likely through mitochondrial ATP production. This study also supports an interesting hypothesis that testis-specific retrogenes compensate for the silencing of X-linked progenitor genes during MSCI. Overall, this manuscript is very well written with insightful discussion. The story about the function of PDHA2 in mitochondria for DSB repair is generally convincing, and the data are of high quality. This paper should be of interest to the readers in the field of germ cell biology and meiosis. Meanwhile, I would like to suggest including additional descriptions of the data, particularly for the nuclear localization and function of PDHA2, which are necessary for supporting their important conclusions.

SUGGESTIONS TO AUTHORS

1. This study used *Pdha2*^{+/-} as a control for *Pdha2*^{-/-}. A description of whether *Pdha2*^{+/-} males are completely normal is needed.

Thank you for the details you pointed out. We ignored the description of this section, but we now supplement the data from *Pdha2*^{+/-} mice in Fig. S3A. Lines 140-141.

2. Supplementary Figure 1C: A TOM20-only panel should also be included to show mitochondrial signals independently.

We supplement the TOM20-only panel as suggested (Fig. S1C).

3. Page 8, lines 164-176: The text appears to be jumbled, likely due to a file error.

We revised the error as suggested. Lines 173-187.

4. Page 6, line 129 states, "when we housed *Pdha2*^{+/-} and *Pdha2*^{-/-} males with three WT females for 2 months separately, no pups were sired from three independent males." Pups were obtained from control males (Fig. 2D) - please correct this mistake. Figure 2D is labeled +/+ and -/-. Please confirm which genotype was used as the control.

We corrected our writing mistakes. Thank you for your reminder. Lines 138-140.

5. Although nuclear signals of PDHA2 are very weak in section IF images, the authors suggest nuclear localization of PDHA2 based merely on spread IF results showing strong signals. Careful evaluation is needed to draw such a conclusion because cytoplasmic proteins can come into contact with nuclear components upon disruption of cellular membranes in spread experiments. Can the authors exclude the possibility that the signals are derived from cytoplasmic PDHA2?

Thank you very much for your critical comments. The nucleus expression of PDHA2 has been reported, and it was supposed to function as an RNA-binding protein (Li et al., 2024). Additionally, because the mitochondrial signals are strong, if we expose more to try to observe nuclear expression in a testis sample it will cause overexposure for mitochondrial signals. Then the IF staining after chromosome spreading is essential for our purpose. To elucidate more reasonable, we supplemented the information on reported nucleus expression in our results. Line 92, 124.

Li, Y., Wang, Y., Tan, Y. Q., Yue, Q., Guo, Y., Yan, R., Meng, L., Zhai, H., Tong, L., Yuan, Z. et al. (2024a) 'The landscape of RNA binding proteins in mammalian spermatogenesis', *Science* 386(6720): eadj8172.

6. Although PDHA1 is localized in XY bodies in the nucleus, the spread IF results suggest that PDHA2 is not associated with XY bodies. This indicates that their roles in the nucleus may be distinct. Since PDHA1 can compensate for the loss of PDHA2 during spermatogenesis, it can be inferred that the primary role of PDHA2 in meiosis is related to mitochondria. This point should be included in the discussion.

Thank you very much for your comments. We strongly agree with your suggestion. We supplemented this part in our discussion. Meanwhile, even though our results supported the essential mitochondrial function of PDHA2, we still cannot ignore the nuclear function. And the function of PDHA1 and PDHA2 in the nucleus may be distinct. This part also should be pointed out. A recent paper suggested PDHA2 may involve in Histon modification in nucleus, our localization data will support their results (Wang et al., 2025). Lines 395-398, 409-415.

Wang G, Fang K, Shang Y, Zhou X, Shao Q, Li S, Wang P, Chen CD, Zhang L, Wang S. Testis-Specific PDHA2 Is Required for Proper Meiotic Recombination and Chromosome Organisation During Spermatogenesis. *Cell Prolif.* 2025 Feb 20:e70003. doi: 10.1111/cpr.70003. Epub ahead of print. PMID: 39973374.

7. Page 10, line 218: Please indicate what tissues/cells from which stage of mice were used for the IP experiment.

We supplemented the information according to your suggestion. Line 232.

8. How the authors identified XY bodies in spread IF experiments should be explained in the Methods and/or legends.

Thank you for your reminder to make it more readable for others. We add this description in the Methods of IF. Lines 876-877.

9. Fig. 5B: Symbols in the *Pdha2* KO column should be explained in the legend.

Thank you for your reminder. This may be because of the leftover from the previous iteration, where Fig. S6C was included alongside. I deleted in case of misunderstanding (Fig. 5B).

10. It may be useful to include gene ontology analysis for the proteins identified from the MS-spec analysis.

We did the GO analysis (Fig. S6B) by using the potential interactomes, which are shown in Fig. 5 B. This data suggested that PDHA2 is mainly involved in pyruvate metabolism and linked to the TCA cycle and Acetyl-CoA, indicating the essential role in testicular mitochondria. Lines 243-245, 825-832.

11. Because the major highlights of this paper are the localization and function of PDHA2 in mitochondria and the nucleus for meiosis, they should include a description of what nuclear proteins were identified from the MS-spec analysis, if any, and provide a discussion. Were RNA-binding proteins included?

Thank you for your suggestion. Because PDHA2 was suggested to be a Histon modification regulator in nucleus (Wang et al., 2025). We supplemented the discussion of interaction candidate proteins, which may be related to epigenetic things, such as HDAC6, PDHX and TARDBP. Lines 235-242, 398-400.

Wang G, Fang K, Shang Y, Zhou X, Shao Q, Li S, Wang P, Chen CD, Zhang L, Wang S. Testis-Specific PDHA2 Is Required for Proper Meiotic Recombination and Chromosome Organisation During Spermatogenesis. *Cell Prolif.* 2025 Feb 20:e70003. doi: 10.1111/cpr.70003. Epub ahead of print. PMID: 39973374.

12. Throughout the manuscript, exact P-values should be shown in figures rather than asterisks/ranges.

Thank you for your suggestion. We supplemented all the exact P-value data in figures on the upper side of asterisks.

Reviewer 2: SUMMARY OF THE ADVANCE MADE IN THIS PAPER AND ITS POTENTIAL SIGNIFICANCE TO THE FIELD

This work concerns the testis-specific peripheral mitochondrial protein Pdha2 (pyruvate dehydrogenase E1 alpha 2). It is postulated that Pdha2 on mouse chromosome 3 is an autosomal retrogene of Pdha1, which is X-linked. Deletion of Pdha2 results in male infertility (azoospermia) due to failure in spermatogenesis at the late pachytene-diplotene transition of meiosis. Interestingly, the phenotype can be rescued by expression of Pdha1 from an autosome (the X-linked Pdha1 would be silenced during meiotic sex chromosome inactivation (MSCI)). Results suggest that PDHA2 forms the pyruvate dehydrogenase complex in mitochondria by interacting with PDHA1 and PDHB, and this is required for normal levels of ATP (for function of ATPase recombinase proteins RAD51 and DMC1) in male germ cells. As RAD51 and DMC1 direct formation of crossovers, their loss of function is associated with defective double-strand break repair during pachytene; the defective germ cells are removed by apoptosis.

A constitutive Pdha2 KO was made, the validity of this was demonstrated, and the phenotype was thoroughly and carefully analysed, creating large amount of data. Although a similar study has very recently been published (Wang et al, *Cell Proliferation*, Feb 2025) the manuscript under consideration is still an important contribution. One part of the current study that adds quite a lot, in my opinion, is that the authors were able to demonstrate, in vivo, that germ cell-specific PDHA2 can be compensated by PDHA1. PDHA1 is used in most cell types, but can not be produced once germ cells enter meiosis, due to MSCI. This study is a good complementary study to the Wang et al, 2005 study mentioned above - together they substantially expand our understanding of the role of the pyruvate dehydrogenase complex (PDC; links glycolysis to the oxidative TCA cycle) in spermatogenesis. The work demonstrates an excellent analysis of the progression of spermatogenesis, with a broad range of characteristic and relevant antibodies used. The complicated process and the roles of the various proteins visualized are well explained. I imagine others will use this work as a template as they analyse mouse models with arrest at a similar stage.

SUGGESTIONS TO AUTHORS

Some questions for the authors to address:

1. I was surprised that PDHA2 interacts physically with PDHA1. It seems more likely that PDHA2 is just required in the special circumstances of MSCI, to compensate for PDHA1 (and complex with PDHB). I think this result might reflect that PDHA proteins normally interact with PDHB to form a heterotetramer (line 79) - so PDHA1 and PDHA2 can interact, though this would not normally be

seen *in vivo*, because they would not normally be co-expressed? Maybe the authors could clarify this point. If I am curious then I presume other readers would be too.

Thank you very much for your critical thinking. We realized that we lacked information and discussion about this part. We introduced more information about the structure of the PDC E1 component, which consists of double PDHB and double PDHA1 in somatic cells. Lines 84-89. Given that *Pdha2* is predicted as a retrogene derived from X-linked *Pdha1*, we propose that the composition of the E1 component of PDC undergoes a dynamic shift during spermatogenesis. We speculate that, in male germ cells, PDHA2 may gradually replace PDHA1 to form a testis-specific E1 configuration. This transition likely proceeds through an intermediate form containing PDHA1, PDHA2, and PDHB (Fig. 5C, Fig. S6D, Fig. S7E), eventually yielding a complex composed of two PDHA2 and two PDHB subunits. We have clarified this point according to your suggestion. Lines 259-268, 366-372.

2. Why would the amount of PDHB decrease when there is no PDHA2? Could it be that it is not stable in the absence of a complexing partner?

Thank you for your questions. We apologize that we didn't mention this point. You are correct. Due to the absence of PDHA2, the PDC component was destroyed, then PDHB became unstable, thus causing decreased amounts. We added this part to the manuscript. Lines 275-276.

3. I wonder whether the title should reflect more about the evolutionary gene pair and what has been revealed regarding functional compensation during MSCI

We modified title as suggested.

4. Line 308 - 'remarkable reductions in ATPase proteins (RAD51 and DMC1) expression are observed'. Is this a reduction in expression/amount of protein though, or a failure to localize? We would probably expect that the protein is made normally, but does not localize to foci, so is not clearly observable?

Thank you very much for your critical suggestion. You are correct. We performed the WB analysis and found protein amounts (RAD51, DMC1, MLH1) are comparable between control and KO mice (PND 21). In some parts we previously used "amount" in the result descriptions for IF, but now we have corrected it. We supplemented the data and discussion at Lines 206-207, 209-210, 374-377, Fig. S5D.

Additional minor points for the authors to address:

1. I had no idea what a 'peripheral' mitochondrial protein is, and it is likely others will also not know.

We supplemented the explanation of peripheral protein. Lines 295, 360.

2. Line 68 - sentence beginning here is not complete.

We revised the sentence as suggested. Lines 74-76.

3. Line 107 - it would be useful to clarify what PA and 1D4 are (as not all readers would be making constructs). This could be in the appropriate figure legends or in the methods.

We supplemented PA and 1D4 tag explanations in the methods as suggested. Lines 718-721.

4. Line 158 - 'MSCI is initiated by BRCA1 localization in early pachytene spermatocytes' - please include a reference here.

We included a reference as suggested. Line 171.

5. Line 164 - from here numerous errors in the text.

We revised the sentence as suggested. Lines from 173.

6. Line 278 - 'testis-specific *Ctgn* promoter' - please include a reference here.

We included a reference as suggested. Line 333.

7. Figure 1 legend. Please include info that Fig1B shows PND testis samples (you just show numbers). Mention in the legend that COX IV is highlighting mitochondria. What is the dotted white line?

We supplemented the description as suggested. We supplemented PND in Fig.1B, COX IV is highlighting mitochondria Line 934, dotted white line 938.

8. Figure 3 legend. Please define DDR and MSCI.

We revised the sentence as suggested. Lines 980-981, 983.

9. Figure 6 legend. 'compensates' in the title

We revised the words as suggested. Line 1040.

10. Supp figure 1 legend. Maybe indicate TOM20 is a mitochondrial marker

We supplemented the description as suggested. Fig. S1.

11. Figure 2. Maybe include the word 'deletion' alongside -2,374 bp

We supplemented the description as suggested. Fig.2 A.

12. Figure 4 . Typo of 'chckpoint'

We revised the words as suggested. Fig.4 E.

13. Figure 5. What is the legend (different colours and shapes for PDHB, PDHA2 etc) in figure 5B referring to? Presumably these are left over from previous iteration where Supp Fig6B was included alongside.

Thank you for your reminder. It's left over from the previous iteration of Fig. S6C. We deleted the different colors and shapes.

14. Figure 6G. Is this labelled correctly (PDHA2 instead of PDHA1 in the upper control panel).

Thanks for your correction, we should write it as PDHA2, we have revised it. Fig. 6G.

Reviewer 3: SUMMARY OF THE ADVANCE MADE IN THIS PAPER AND ITS POTENTIAL SIGNIFICANCE TO THE FIELD

Pan et al present a detailed characterization of the meiotic arrest associated with a loss of PDHA2, a conserved retrogene copied from the X-linked gene PDHA1. PDHA1 is an essential component of the mitochondrial pyruvate dehydrogenase complex (PDC) that is inactivated by MSCI in spermatocytes. The authors show that in mice lacking PDHA2 there is a block in late pachytene prior to diplotene resulting from a failure to correctly repair double strand breaks (DSB). They further show that PDHA2 interacts with PDHA1 and PDHB, components of the PDC, and that ectopic PDHA1 expression in spermatocytes complements for the loss of PDHA2, evidence that PDHA2 functions similarly to PDHA1. In spermatocytes lacking PDHA2, they find reduced levels of ATP and a reduction of DNA repair ATPase proteins RAD51 and DMC1, as well as MLH1 foci (crossovers) on the chromosomes in mid-late pachytene spermatocytes. Finally, the authors present evidence that PDHA1 and PDHA2 localise to the nucleus in pachytene spermatocytes. Although determining the functional significance of this observation will not be easy, it could inspire new avenues of exploration.

The study has been thoroughly performed and the images presented are convincing. An interesting story has been told, in particular around the functional complementarity of PDHA1 and its retrogene PDHA2 - based on both a genetic and proteomics approach. Nevertheless, the article is harder to understand than it need be for someone who does not regularly attend the authors' lab meetings. I have indicated areas that were problematical for me, but the authors should make an effort to revise the text to render it as readable as possible. Despite the publication two weeks ago of an independent article describing the effects of PDHA2-loss on meiotic progression, the present article would make a significant contribution to the field since it adds clean informative proteomics data and a demonstration that PDHA1 complements the loss of PDHA2. A comparison of perceived phenotype differences in Pdha2-KO mice may also be illuminating.

SUGGESTIONS TO AUTHORS

The finding that PDHA1 and 2 localise to the nucleus is unexpected for mitochondrial proteins (could the authors add a clear statement as to whether this has already been described or not) especially since their partner PDHB does not. I am not challenging this result, which looks convincing for PDHA1, but the authors should make more effort to show this to be a secure finding for PDHA2. Ideally the specificity and affinity of the antibody raised by the authors for this study against PDHA2 must be validated for the meiotic spreads. For the anti-PDHA2, this should be done on PDHA2+/-, PDHA2-/- + C1gn-PDHA1-tg, and PDHA2-/- mice. This would also show whether there

is any cross-reaction with PDHA1. Furthermore, Figure 5 seems to show nicely that PDHA1 signal colocalizes with gamma-H2AX (XY body), but this is not evident in supp Figure 2A for PDHA2. This must be discussed.

We sincerely thank the reviewer for the insightful and constructive comments, which have greatly helped us improve the quality and clarity of our manuscript.

First, regarding the nuclear localization of PDHA2, we have added a clear statement in the manuscript (Lines 92,124) and cited a recent report by Li et al. (2024), which previously described PDHA2 expression to the nucleus. This supports our unexpected but reproducible observation. Secondly, to validate the specificity and affinity of the self-generated antibodies (anti PDHA1, PDHA2, and PDHB) used in this study, we have now included the full IP-WB membranes in the revised Fig. S 6D. These results clearly demonstrate the reliable performance of all three antibodies. In addition to the original WB data shown in Fig. S 2C, 6C, and 6E, where *Pdha2*^{+/-}, *Pdha2*^{-/-}, and *Pdha2*^{-/-} + *C1gn-Pdha1* transgenic testes were analyzed, we further addressed the concern regarding potential cross-reactivity. Specifically, we performed additional validations using liver protein extracts (which express PDHA1 but not PDHA2) and HEK293T cells transfected with PDHA1 or PDHA2 expression constructs. These *in vivo* and *in vitro* assays consistently confirmed that the antibodies are highly specific and do not cross-react (Fig. S7D, E; Lines 250-259).

Third, as the reviewer correctly noted, PDHA1 shows strong colocalization with γ H2AX in the XY body (Fig. 5F, Fig. S8C), whereas PDHA2 does not show a similar pattern (Fig. 1D). This distinction is now clearly discussed in the revised Discussion section (Lines 409-415), where we highlight that although both proteins can localize to the nucleus, their roles may be distinct. Since PDHA1 (not PDHA2) is enriched at the XY body and can compensate for PDHA2 deficiency during spermatogenesis, we propose that the primary function of PDHA2 is mitochondrial, whereas PDHA1 may have an additional role in the regulation of sex chromosome-associated processes during meiosis.

Li, Y., Wang, Y., Tan, Y. Q., Yue, Q., Guo, Y., Yan, R., Meng, L., Zhai, H., Tong, L., Yuan, Z. et al. (2024a) 'The landscape of RNA binding proteins in mammalian spermatogenesis', *Science* 386(6720): eadj8172.

On a related point, there is some confusion in the labelling of Figure 6G. The upper central and upper right panel do not accord in their PDHA labelling - is the magenta PDHA1 or PDHA2?

Thank you for your correction, Due to carelessness, we confused the mark; now that we have corrected it, PDHA2 instead of PDHA1 in the upper control panel.

I was initially confused by the finding that PDHA1 is a potential partner of PDHA2. This suggested either that normally PDHA1 and PDHA2 can be part of the same PDC in spermatocytes, or that each of their anti-PDHA antibodies detects both PDHA1 and PDHA2. I discovered, by literature search, that PDHA1 appears to form homodimers, and that its heterotetramer with PDHB, to give the pyruvate dehydrogenase E1, exists in multiple copies within each PDC. This information is extremely important for the interpretation of this study and it will greatly improve the understanding of this article and the perceived quality of the results for scientists who are not familiar with the PDC. The structure of the PDC must be more fully presented in the Introduction, or at least discussed in the context of the results.

Thank you for your critical comments to improve the quality of our manuscript. We added the description of PDC E1 structure in the introduction and context of the results, which is consistent with two PDHA1 and two PDHB in somatic cells (Lines 84-89). This heterotetramer structure exists in multiple copies within each PDC. Thus, we discussed again why the interaction of PDHA1 and PDHA2 could be found in the testis. They may have a transit heterotetramer structure that contains a PDHA1, a PDHA2, and two PDHB during spermatogenesis and transformation. Lines 259-268.

Antibody validation. The authors have generated three antibodies against three of the key proteins in this study, PDHA1, PDHA2 and PDHB. Information must be provided in Material and Methods concerning the strategy for immunogen choice (especially for PDHA1 and 2), immunogens, affinity validation and demonstration of a lack of cross-reaction with PDHA1 or PDHA2. The validation of the anti-PDHA2 specificity in denaturing conditions has been convincingly done by the Western analyses of the *Pdha2*-KO mice. At face value, the IP results suggest that there is equally little cross-reaction between PDHA1 and 2 under non-denaturing conditions. The IF in Supp Fig 2B shows that the anti-PDHA2 certainly has low affinity for PDHA1. A full length Western for each of the three new antibodies should be shown in a supp Figure.

Thank you for your cautious reminder. Many results rely on our self-made antibodies in this project. As PDHA1 and PDHA2 show 95% similarity in amino acids (Fig. S7A), it's important to do antibody validation. To check it, we performed an *in vitro* expression analysis using *Pdha1* and *Pdha2* expression vectors in HEK cells. We then found that both antibodies could only recognize each target protein (Fig. S7D). We also extracted liver proteins *in vivo* and performed IP-WB analysis (Fig. S7E). We confirmed that PDHA2 only showed the band in the testis sample but not in the liver. On the other hand, PDHA1 expresses in both the testis and liver. IP analysis using anti-PDHA1, anti-PDHA2 also revealed that both antibodies recognize their target proteins. In summary, there is no cross-reaction between anti-PDHA1 and PDHA2 antibodies Lines 250-259. We added a protocol for antibody generation to the materials and methods as suggested (Lines 754-759, 763-765). A full-length Western blot analysis for each of the three self-made antibodies was added as shown in Fig. S6D.

Please state clearly and as precisely as possible the mouse strain or hybrid used in the two different sets of transgenic experiments, taking into account the breeding strategy used to generate the different PDHA genotypes studied. The only information provided is "B6D2F1 (C57BL/6 × DBA2), ICR, and C57BL6/J mouse strains were utilized for embryo donation, fostering, and RNA extraction, respectively." at line 563.

We supplemented the description as suggested. Lines 652, 654, 655-656, 662, 665-666.

Please present the criteria used to classify meiotic spreads as mid- or late-pachytene.

Thank you so much for your question. To unambiguously distinguish mid- and late-pachytene spermatocytes, we employed a multi-tiered approach combining histone markers, chromosomal morphology, and stage-specific cytological features. The classification hierarchy is as follows: First, we used H1t to distinguish the middle and late pachytene compared to early pachytene spermatocytes. H1t signal will show from middle pachytene and increase in late pachytene compared to middle. This is a sample first step, but not a very accurate method because exposure time or the condition of the confocal microscope may affect the result. Next, based on the H1t signal, we distinguished stages by the sex chromosome morphology and autosome length/shape by SYCP3 staining. On the one hand, the sex chromosomes during the middle pachytene stage will align together without a loop, but will show in the late pachytene stage. On the other hand, the KO mice may relate to an abnormal XY body, such as *Pdha2*. We distinguish by SYCP3 staining, weak SYCP3 signal enrichment at chromosomes' termini with shorter, thicker autosomal axes at the middle pachytene stage compared to the late pachytene stage. Strong SYCP3 accumulation on telomeric regions could be observed on late pachytene autosomes. Taken together, by these three methods, we can be sure to classify the middle-late pachytene spermatocytes. We supplemented this information in text. Lines 877-880.

Alavattam KG, Abe H, Sakashita A, Namekawa SH. Chromosome Spread Analyses of Meiotic Sex Chromosome Inactivation. *Methods Mol Biol.* 2018;1861:113-129. PMID: 30218364

Please discuss the independent article that has just been published (Fang et al 2025) describing the phenotype associated with loss of PDHA2 function in the mouse. In particular the difference in interpretation of the point of arrest (late pachytene vs diplotene). If possible, state whether or not abnormal diplotene cells are present in the model presented here.

Thank you for your comments. Prior to our study, three publications had reported the phenotype of *Pdha2* knockout (KO) mice: Fang et al., 2023, Y. Li et al., 2024, and Wang G. et al., 2025. All three studies suggest disruption during the transition from the pachytene to diplotene stages in meiosis. Li et al., 2024 reported that *Pdha2* KO spermatocytes were arrested at the pachytene stage, potentially due to the dysregulation of the RNA-binding processes. More recently, Wang et al. (2025) proposed a role for PDHA2 in histone modifications, likely linked to reduced nuclear acetyl-CoA levels observed in KO spermatocytes. In contrast, Fang et al. (2025) interpreted the phenotype as stagnation at a diplotene-like stage, whereas our study concludes that arrest occurs in late pachytene. We did observe a small number of abnormal cells in the KO spermatocytes as discussed, notably in RAD51 immunofluorescence staining. However, due to too many abnormalities in these cells, it was not feasible to perform a reliable quantification. Despite these differences in staging criteria, **both studies consistently demonstrated defects in the pachytene-to-diplotene transition in *Pdha2* KO mice.** Both studies also reported a **reduction in crossover formation** in KO pachytene spermatocytes. Thus, the apparent discrepancy in staging might be more a matter of

interpretative emphasis than a fundamental disagreement. We supplement these things in the discussion. Lines: 395-398.

In accordance with the journal's requirements, we have further provided detailed information on the image selection and quantification criteria. We mentioned about the cells displaying abnormal morphology, they were excluded from the statistical analysis. Lines 888-901.

Please state in the results that spermatocytes were enriched by flow sorting prior to measuring ATP levels. This information should be added to the Methods section. An estimate of purity should be added.

We apologize that our manuscript caused you a misunderstanding. We didn't use FACS to sort the cells to check the ATP amounts. In this experiment, we only did a single cell suspension using PND 21 mice. We used a 40 µm mesh to filter the cells, which can remove other cells except stem cells due to their size. Then we collected the filtered cells for the ATP amount essay. Lines 791-800.

Bastos H, Lassalle B, Chicheportiche A, Riou L, Testart J, Allemand I, Fouchet P. Flow cytometric characterization of viable meiotic and postmeiotic cells by Hoechst 33342 in mouse spermatogenesis. *Cytometry A*. 2005 May;65(1):40-9. PMID: 15779065.

The mitochondrial localization experiment presented in Figure 5G and H is poorly explained in the results - lines 252-258.

- "but transmembrane proteins, [such as VDAC1], appear in the pellet."

- "This treatment changes the localization of" is not clear. You could state the observed change.

- "immunoblot signals comparable to those of UQCRC1...." A little introduction to UQCRC1, a protein attached to the inner membrane, is required here.

We supplemented the explanation of this part according to your suggestion. Lines 293-304.

Minor comments

In light of Fang et al 2025, I would recommend re-working the title to add the original finding that PDHA1 can complement for the loss of PDHA2.

We modified the title as suggested.

Line 25: I do not think you can say that PDHA2 serves the same roles as PDHA1 in somatic cells. To do this would require you to complement the loss of PDHA1 with PDHA2. You can however say that PDHA1 is able to perform the same roles as PDHA2 in the mitochondria of spermatocytes. You could move "and is essential for [normal] ATP production in pachytene spermatocytes" to the second highlight.

Thank you for your suggestion. We understand the difference between these two sentences. Due to the format requirements of journal, we deleted the highlights and changed them to Summary Statement. Lines 27-29. And we understand that point, we avoid saying it in our manuscript.

Line 120: >>> Therefore we speculate that PDHA2 may play a role in the nucleus as well as the mitochondria during male meiosis.

We revised the sentence as suggested. Line 129-130.

Line190: [Compared to controls,] there were no differences in HORMAD1 localisation on X and Y

We revised the sentence as suggested. Line 202.

Line 317: "A previous study suggested" >>> Previous studies suggested

We deleted this sentence because we changed the discussion part.

Line 347: "We hope the analysis". The sense of this sentence is not obvious.

We deleted this sentence because we changed the discussion part.

Line 343: "This finding supports the hypothesis.....". No reference is provided. It is more logical to propose that the creation of the PDHA2 retrogene allowed the silencing of PDHA1 by MSCI.

We revised the sentence as suggested. Lines 423-425.

Please add the age of the mice used in the Figures, where space permits. Otherwise make sure this is clearly mentioned in the Figure legend.

We agree that the age of mice used in this project is very important. We supplemented the mice's age in all Figure legends if there was no information in the Figures. Lines 942, 954, 973, 979, 988, 991, 1003, 1006, 1009, 1017, 1022, 1035, 1052. Supplementary Figures.

Reviewer 4: SUMMARY OF THE ADVANCE MADE IN THIS PAPER AND ITS POTENTIAL SIGNIFICANCE TO THE FIELD

This study by Pan et al. systematically investigates the role of testis-specific PDHA2 in male meiosis using mice models. The authors demonstrate that PDHA2 interacts with PDHB in testicular mitochondria to regulate mitochondrial functions. PDHA2 deficiency leads to ATP depletion in germ cells, impairing ATPase-mediated recombination protein functions and causing insufficient crossover formation during pachytene stage. Consequently, spermatocytes arrest at the pachytene stage and undergo apoptosis, ultimately resulting in azoospermia. Transgenic *Pdha1* expression rescued fertility in *Pdha2* KO (knock-out) mice, supporting the evolutionary hypothesis of PDHA2's origin from PDHA1 while explaining the pachytene-specific defects. The study is logically structured with solid evidence, advancing our understanding of mitochondrial proteins in spermatogenesis and offering therapeutic insights for PDHA2-related human infertility.

SUGGESTIONS TO AUTHORS

Major Comments

1. ATP restoration validation: While the ATP depletion mechanism is clearly demonstrated, did the authors attempt to supplement ATP in *Pdha2*-KO testes to recover male fertility?

Thank you for your question. We hypothesized that ATP supplementation in germ cells could rescue the fertility of *Pdha2* KO mice. We supplemented 5 μ m ATP (Grootegeod JA, et al. 1984) in culture medium by using our new *in vitro* spermatogenesis system (Kamoshita M et al. 2025). However, we didn't find any round spermatids in ATP supplemented KO testis, indicating *Pdha2* KO spermatocytes could not pass the meiosis-checkpoint at least. It may be due to the lack of nucleus function of PDHA2, or ATP could not go through the cell membrane. Although we performed this *in vitro* analysis, the experimental method has not yet been fully established and there may be problems with the protocol. Therefore, we did not include this supplemental experiment using an *in vitro* culture system in our paper.

Grootegeod JA, Jansen R, Van der Molen HJ. The role of glucose, pyruvate and lactate in ATP production by rat spermatocytes and spermatids. *Biochim Biophys Acta*. 1984 Nov 26;767(2):248-56. PMID: 6498180.

Kamoshita M, Shirai H, Nakamura H, Kishimoto T, Hatanaka Y, Mashiko D, Esashika K, Yang J, Yamasaki S, Ogawa T, Kimura H, Ikawa M. Development of the membrane ceiling method for *in vitro* spermatogenesis. *Sci Rep*. 2025 Jan 3;15(1):625. PMID: 39753886.

NOTE: We have removed unpublished data that had been provided for the referees in confidence.

2. Metabolic compensation: PDHA2 is essential for mitochondrial function. Were alternative energy substrates (e.g., fatty acids or glutamine) examined for potential compensatory effects on spermatogenesis of this KO?

Thank you for your consideration. We also wondered if glutamine supplementation would rescue the ATP amounts and then rescue the fertility of *Pdha2* KO mice. In our *in vitro* spermatogenesis system, we originally contained a large amount of glutamine. MEM α (Invitrogen, Carlsbad, CA), Advanced DMEM/F12 (Invitrogen, Carlsbad, CA), and AlbuMAX I (ThermoFisherScientific, Waltham, MA) were used. However, we still could not rescue the infertility of our KO mice. We did not include this experiment in our paper either.

NOTE: Figure provided for reviewer has been removed. It showed data from Kamoshita M, Shirai H, Nakamura H, Kishimoto T, Hatanaka Y, Mashiko D, Esashika K, Yang J, Yamasaki S, Ogawa T, Kimura H, Ikawa M. Development of the membrane ceiling method for *in vitro* spermatogenesis. *Sci Rep*. 2025 Jan 3;15(1):625. PMID: 39753886. We have removed unpublished data that had been provided for the referees in confidence.

Kamoshita M, Shirai H, Nakamura H, Kishimoto T, Hatanaka Y, Mashiko D, Esashika K, Yang J, Yamasaki S, Ogawa T, Kimura H, Ikawa M. Development of the membrane ceiling method for in vitro spermatogenesis. *Sci Rep.* 2025 Jan 3;15(1):625. PMID: 39753886.

3. Mitochondrial consequences: Figure 5D/E and Supplementary Figure 2B seems show mitochondrial morphological changes. Were mitochondrial dynamics or oxidative phosphorylation proteins analyzed by WB?

Thank you for your question. It's important to validate other mitochondrial proteins' function in *Pdha2* KO mice. We supplemented the data about comparable TOM 20, COX IV, VDAC1, UQCRC1, and ATP5A protein amounts between control and KO testis (PND21, Fig. S8A). This result indicated that deletion of PDHA2 does not affect other proteins. Due to a lack of metabolome data, we don't know if the function of these proteins is normal or not. Lines 275-277, 364-365.

4. PDHA1/2 specificity:

- The hypothesis that PDHA2 originated from PDHA1 would be strengthened by including a DNA sequence alignment between the two genes, particularly in regulatory regions.

We supplemented this part in the Fig. S9A, B as suggested. Line 318-321.

- Supplementary Fig. 6B/7B indicate that X-linked PDHA1 is silenced during MSCI. Does the *Clgn* promoter-driven transgene presumably expresses PDHA1 post-MSCI? How does the Tg bypass MSCI-mediated silencing?

Clgn promoter (Lu et al., 2023), as a germ-cell specific promoter, will drive *Pdha1* to overexpress strongly in spermatocytes and spermatids. Because *Pdha1* localizes to several autosomes in Tg mice, *Pdha1* can express after MSCI. As a result, PDHA1 replaced the function of PDHA2 in *Pdha2* KO mice after MSCI and rescued their fertility. We also supplemented explanation and reference in manuscript. Line 333.

Lu Y, Shimada K, Tang S, Zhang J, Ogawa Y, Noda T, Shibuya H, Ikawa M. 170002915Rik orchestrates the biosynthesis of acrosomal membrane proteins required for sperm-egg interaction. *Proc Natl Acad Sci U S A.* 2023 Feb 21;120(8):e2207263120. Epub 2023 Feb 14. PMID: 36787362.

5. Nuclear function clarification: Figure 6 and Supplementary Fig. 6 shows that transgenic *Pdha1* restores both mitochondrial and nuclear localization. It seems PDHA1 persists in the nucleus beyond mid-pachytene. Does PDHA2-KO impair nuclear-specific functions? This observation strengthens the rescue mechanism and warrants discussion.

Thank you for your question. A recent paper implicated the nucleus function of PDHA2 in histone modification, likely due to reduced levels of nuclear acetyl-CoA observed in *Pdha2* KO spermatocytes (Wang et al., 2025). And because PDHA2 does not localize to XY bodies, we discussed that PDHA2 and PDHA1 may have distinct functions in the nucleus. We mentioned these things in the discussion. The distinct functions of PDHA2 in the nucleus and mitochondria still need further exploration. We supplemented this part in the discussion. Lines 395-400, 409-415.

Wang G, Fang K, Shang Y, Zhou X, Shao Q, Li S, Wang P, Chen CD, Zhang L, Wang S. Testis-Specific PDHA2 Is Required for Proper Meiotic Recombination and Chromosome Organisation During Spermatogenesis. *Cell Prolif.* 2025 Feb 20:e70003. doi: 10.1111/cpr.70003. Epub ahead of print. PMID: 39973374.

Minor Comments

1. Figure issues:

-Figure 3B/D: Labels appear inconsistent between panels and legends ("WT" vs "Hetero")

-Figure 4E: Missing annotation

We corrected the mistake in Fig. 3B.D, 4E. as suggested.

2. Line 71-73: Rephrase metabolic demand statement to: "This highlights the heightened metabolic requirements of meiotic cells, suggesting energy deficits during this phase critically disrupt spermatogenesis."

We revised the sentence as suggested. Lines 77-79.

3. Line 81-83: Clarify unresolved mechanistic links.

We revised the sentence as suggested. Lines 90-92.

4. Line 130: Replace "three independent males" with "KO mating cages".

We revised the sentence as suggested. Line 139.

5. Line 159, 162, 164, 166, 168, 172, 174, 195: Address irregular spacing and citation formatting (reference insertion error).

We revised the sentence as suggested. Lines 173-187.

6. Line 163: Remove redundant "IF".

We revised the sentence as suggested. Line 176.

7. Line 167-169: Revise to correct multiple format and description errors.

We revised the sentence as suggested. Lines 173-187. We apologize for multiple mistakes here.

8. Line 173: Remove redundant parentheses. And delete incomplete sentence: "During MSCI, the sex chromosomes are transcriptionally."

We revised the sentence as suggested. Lines 179.

9. Line 174-175: Consolidate redundant mentions of H3K9me3.

We revised the sentence as suggested. Lines 182.

10. Line 299-301: Rewrite for clarity.

We revised the sentence as suggested. Lines 349-358, 366-372.

11. Line 578: Delete extraneous "-" after PA.

We revised the sentence as suggested. Line 659.

12. Line 794: Correct formatting error.

We revised the sentence as suggested. Line 939.

13. Line 807: Ensure consistent italicization of "N".

We revised the sentence as suggested. Line 955.

Second decision letter

MS ID#: dev.204683R1

MS TITLE: Compensation for X-linked Pdha1 silencing by Pdha2 is essential for meiotic double-strand break repair in spermatogenesis

AUTHORS: Chen Pan, Keisuke Shimada, Hsin-Yi Chang, Haoting Wang and Masahito Ikawa

Dear Dr Shimada,

I have now received all the referees reports on the above manuscript, and have reached a decision. The referees' comments are appended below, or you can access them online: please go to .

The overall evaluation is positive and we would like to publish a revised manuscript in Development, provided that the referees' comments can be satisfactorily addressed. Please attend to all of the reviewers' comments in your revised manuscript and detail them in your point-by-point response. If you do not agree with any of their criticisms or suggestions explain clearly why this is so. If it would be helpful, you are welcome to contact us to discuss your revision in greater detail. Please send us a point-by-point response indicating your plans for addressing the referees' comments, and we will look over this and provide further guidance.

Reviewer 1*Advance summary and potential significance to field*

The authors have addressed most of my previous concerns, and the manuscript has improved substantially. The evidence demonstrating the crucial role of PDHA2 in double-strand break repair through energy production is convincing, and its compensation for PDHA1 is intriguing. I have one additional comment that should be further considered.

Comments for the author

The evidence supporting PDHA2 localization in the nucleus has been strengthened by the inclusion of a previous report, which further supports the signals detected in the spread IF experiment (Fig. 1D). However, its localization in COS-7 cells remains unconvincing due to the presence of nuclear signals in the mock condition and the overlap of faint nuclear PDHA2 with TOM20. The statement 'PDHA2 co-localized with the mitochondrial marker TOM20, and localized in the nucleus with weak signals (line 119-120)' should be softened.

Reviewer 2*Advance summary and potential significance to field*

The authors have carefully and thoroughly addressed all four reviewers questions and comments, in my opinion and to my satisfaction. This will be a paper of great interest to those in interested in germ cell biology and meiotic mechanisms.

Reviewer 3*Advance summary and potential significance to field*

1) PDHA1 can complement the loss of PDHA2 function during spermatogenesis.
 2) PDHA2 is required for mitochondrial pyruvate dehydrogenase activity during meiosis when the X-linked PDHA1 is inactivated by MSCI.
 3) Links mitochondrial ATP level regulation and DSB repair during meiotic recombination. Further understanding of a process critical to meiosis. A new example of a retrogene that can compensate for the loss of its X-linked paralogue through MSCI in spermatocytes, providing new insights into sex chromosome function and evolution.

Comments for the author

The authors have responded admirably to my comments. The article is greatly improved. In the summary statement: Given the high degree of nucleotide identity between PDHA1 and PDHA2, "likely derived from PDHA1" seems unnecessarily cautious >>> "a retrogene derived from PDHA1" may be better, unless there are other serious candidate genes.

Reviewer 4*Advance summary and potential significance to field*

The authors may consider to revise the structure of title to more concisely reflect the main findings.

Second revisionAuthor response to reviewers' comments**General comments:**

We sincerely thank you for your valuable comments, which have significantly improved our manuscript. We have revised the manuscript as suggestions. Below is a response to reviewers' comments with the original comments in black and our answers in blue. Following the editor's instructions, we have not used the "Track Changes" function; instead, all textual revisions are marked in red.

Response to the reviewers:

Reviewer 1: SUMMARY OF THE ADVANCE MADE IN THIS PAPER AND ITS POTENTIAL SIGNIFICANCE TO THE FIELD

The authors have addressed most of my previous concerns, and the manuscript has improved substantially. The evidence demonstrating the crucial role of PDHA2 in double-strand break repair through energy production is convincing, and its compensation for PDHA1 is intriguing. I have one additional comment that should be further considered.

SUGGESTIONS TO AUTHORS

The evidence supporting PDHA2 localization in the nucleus has been strengthened by the inclusion of a previous report, which further supports the signals detected in the spread IF experiment (Fig. 1D). However, its localization in COS-7 cells remains unconvincing due to the presence of nuclear signals in the mock condition and the overlap of faint nuclear PDHA2 with TOM20. The statement 'PDHA2 co-localized with the mitochondrial marker TOM20, and localized in the nucleus with weak signals (line 119-120)' should be softened.

Thank you so much for your correction. Because the localization also may different between COS-7 cells and testis, so we erased this sentence (localized in the nucleus with weak signals) according to your suggestion. Line: 119

Reviewer 2: SUMMARY OF THE ADVANCE MADE IN THIS PAPER AND ITS POTENTIAL SIGNIFICANCE TO THE FIELD

The authors have carefully and thoroughly addressed all four reviewers questions and comments, in my opinion and to my satisfaction.

This will be a paper of great interest to those in interested in germ cell biology and meiotic mechanisms.

Thank you so much for your help, your suggestions have made our manuscript better.

SUGGESTIONS TO AUTHORS

Reviewer 3: SUMMARY OF THE ADVANCE MADE IN THIS PAPER AND ITS POTENTIAL SIGNIFICANCE TO THE FIELD

1) PDHA1 can complement the loss of PDHA2 function during spermatogenesis.

2) PDHA2 is required for mitochondrial pyruvate dehydrogenase activity during meiosis when the X-linked PDHA1 is inactivated by MSCI.

3) Links mitochondrial ATP level regulation and DSB repair during meiotic recombination.

Further understanding of a process critical to meiosis. A new example of a retrogene that can compensate for the loss of its X-linked paralogue through MSCI in spermatocytes, providing new insights into sex chromosome function and evolution.

SUGGESTIONS TO AUTHORS

The authors have responded admirably to my comments. The article is greatly improved.

In the summary statement: Given the high degree of nucleotide identity between PDHA1 and PDHA2, "likely derived from PDHA1" seems unnecessarily cautious >>> "a retrogene derived from PDHA1" may be better, unless there are other serious candidate genes.

Thank you so much for your questions, we appreciate that your suggestions have made our manuscript better. And we have revised the summary statement as suggested. Line:28

Reviewer 4: The authors may consider to revise the structure of title to more concisely reflect the main findings.

Thank you so much for your suggestion, we have revised the title as suggested.

Third decision letter

MS ID#: dev.204683R2

MS TITLE: Compensation for X-linked Pdha1 silencing by Pdha2 is essential for meiotic double-strand break repair in spermatogenesis

AUTHORS: Chen Pan, Keisuke Shimada, Hsin-Yi Chang, Haoting Wang and Masahito Ikawa

Dear Dr Shimada,

I am happy to tell you that your manuscript has been accepted for publication in Development, pending our standard publication integrity checks.